# Perception and memory have distinct spatial tuning properties in human visual cortex

**Serra E. Favila** [1,5] ✉, **Brice A. Kuhl** [2,3] **& Jonathan Winawer** [1,4]

Reactivation of earlier perceptual activity is thought to underlie long-term memory recall. Despite evidence for this view, it is unclear whether mnemonic activity exhibits the same tuning properties as feedforward perceptual activity. Here, we leverage population receptive field models to parameterize fMRI activity in human visual cortex during spatial memory retrieval. Though retinotopic organization is present during both perception and memory, large systematic differences in tuning are also evident. Whereas there is a three-fold decline in spatial precision from early to late visual areas during perception, this pattern is not observed during memory retrieval. This difference cannot be explained by reduced signal-to-noise or poor performance on memory trials. Instead, by simulating top-down activity in a network model of cortex, we demonstrate that this property is well explained by the hierarchical structure of the visual system. Together, modeling and empirical results suggest that computational constraints imposed by visual system architecture limit the fidelity of memory reactivation in sensory cortex.

Episodic memory retrieval allows humans to bring to mind the details of a previous experience. This process is hypothesized to involve regenerating sensory activity that was evoked during the initial event[1–4]. For example, remembering a friend's face might invoke neural activity that was present when seeing that face. There is considerable evidence from human neuroimaging demonstrating that the same visual cortical areas active during perception are also active during imagery and long-term memory retrieval[5–14]. These studies have found that mnemonic activity in early visual areas like V1 reflects the low-level visual features of remembered stimuli, such as spatial location and orientation[5,12,15–17]. Likewise, category-selective activity in high-level visual areas like FFA and PPA is observed when participants remember or imagine faces and houses[6,7,10]. The strength and pattern of visual cortex activity has been associated with retrieval success in memory tasks[11,18–20], suggesting that cortical reactivation is relevant for behavior.

These studies, and others performed in animal models[21], have established similarities between the neural substrates of visual perception and visual memory. However, relatively less attention has been paid to identifying and explaining differences between activity patterns evoked during perception and memory. Which properties of stimulus-driven activity are reproduced in visual cortex during memory retrieval and which are not? The extreme possibility—that all neurons in the visual system produce identical responses when perceiving and remembering a given stimulus—can likely be rejected. Early studies demonstrated that sensory responses were reduced during memory retrieval relative to perception[7,8], and perception and memory give rise to distinct subjective experiences. A second, and more plausible, proposal is that visual memory functions as a weak version of feedforward perception, with memory activity in visual cortex organized in the same fundamental way as perceptual activity[22], but with reduced signal-to-noise[23,24] caused by weak or imperfect inputs from the memory system. This hypothesis is consistent with informal comparisons between perception and memory BOLD amplitudes and data suggesting that visual imagery produces similar behavioral effects to weak physical stimuli in many tasks[25–28]. A third possibility is that memory reactivation differs from stimulus-driven activation in predictable and systematic ways beyond signal-to-noise. Such differences could arise due to a change in the neural populations recruited, a change in those populations' response properties, or a systematic loss

[1]Department of Psychology, New York University, New York, NY 10003, USA. [2]Department of Psychology, University of Oregon, Eugene, OR 97403, USA. [3]Institute of Neuroscience, University of Oregon, Eugene, OR 97403, USA. [4]Center for Neural Science, New York University, New York, NY 10003, USA. [5]Present address: Department of Psychology, Columbia University, New York, NY 10027, USA. ✉e-mail: sef2177@columbia.edu

of information during sensory encoding. Critically, any systematic differences between visual cortical response properties during perception and memory should be observable under conditions of high memory fidelity, when memory strength and accuracy are maximized.

One way to adjudicate between these possibilities is to make use of encoding models from visual neuroscience that quantitatively parameterize the relationship between stimulus properties and the BOLD response. In the spatial domain, population receptive field (pRF) models define a 2D receptive field that transforms stimulus position on the retina to a voxel's BOLD response[29,30]. These models are based on well-understood physiological properties of the primate visual system and account for a large amount of variance in the BOLD signal observed in human visual cortex during perception[31]. Using these models to quantify memory-evoked activity in the visual system offers

the opportunity to precisely model the properties of memory reactivation in visual cortex and their relationship to the properties of visual activation. In particular, the fact that pRF models describe neural activity in terms of stimulus properties may aid in interpreting differences between perception and memory activity patterns by projecting these differences onto a small number of interpretable physical dimensions. Such approaches have already proved effective in answering some questions about the nature of visual working memory representations. Numerous studies have used encoding models to show that visual cortex encodes stimuli maintained in working memory in a similar format to perception[32–35]. However, these studies have emphasized similarity between working memory and perception and have not reported systematic differences between the two. Moreover, there are good theoretical reasons to suspect that results from working memory studies may not generalize to episodic memory. Namely, while working memory is thought to depend on the maintenance of perceptual activity that was evoked seconds ago, episodic memory retrieval requires the total reinstantiation of perceptual activity that was evoked minutes, hours, or days ago. These different cognitive operations may impose different constraints on stimulus representations.

In this work, we use pRF models to characterize the spatial tuning properties of mnemonic activity in human visual cortex during episodic memory recall. We first trained human participants to associate spatially localized stimuli with colored fixation dot cues. We then measured stimulus-evoked and memory-evoked activity in visual cortex using fMRI. Separately, we fit pRF models to independent fMRI data, allowing us to estimate receptive field locations within multiple visual areas for each participant. Using pRF-based analyses, we quantify the location, amplitude, and precision of neural activity within these visual field maps during perception and memory retrieval. We find fundamental differences in the amplitude and precision of perceptual and mnemonic activity that span the visual hierarchy and that cannot be explained by low signal-to-noise or memory failure. Finally, we explore the cortical computations that could account for our observations by simulating neural responses using a stimulus-referred pRF model and a hierarchical model of neocortex. We find that basic patterns in our empirical results can be accounted for by reversing the flow of information between perception and memory retrieval within a hierarchical model of cortex.

## Results
### Behavior

Prior to being scanned, participants ($N = 9$) completed a behavioral training session. During this session, participants learned to associate four colored fixation dot cues with four stimuli. The four stimuli were unique radial frequency patterns presented at 45, 135, 225, or 315 degrees of polar angle and 2 degrees of eccentricity (Fig. 1a, b). Participants alternated between study and test blocks (Fig. 1c). During study blocks, participants were presented with the associations. During test blocks, participants were presented with the fixation dot cues and had to detect the associated stimulus pattern and polar angle location among similar lures (Fig. 1a, c; see Methods). All participants completed a minimum of four test blocks (mean = 4.33, range = 4–5), and continued the task until they reached 95% accuracy. Participants' overall performance improved over the course of training session (Fig. 1d). In particular, participants showed improvements in the ability to reject similar lures from the first to the last test block (Fig. 1e).

After participants completed the behavioral training session, we collected fMRI data while participants viewed and recalled the stimuli (Fig. 2a). During fMRI perception runs, participants fixated on the central fixation dot cues and viewed the four stimuli in their learned spatial locations. Participants performed a one-back task to encourage covert attention to the stimuli. Participants were highly accurate at detecting repeated stimuli (mean = 86.9%, range = 79.4%–93.2%).

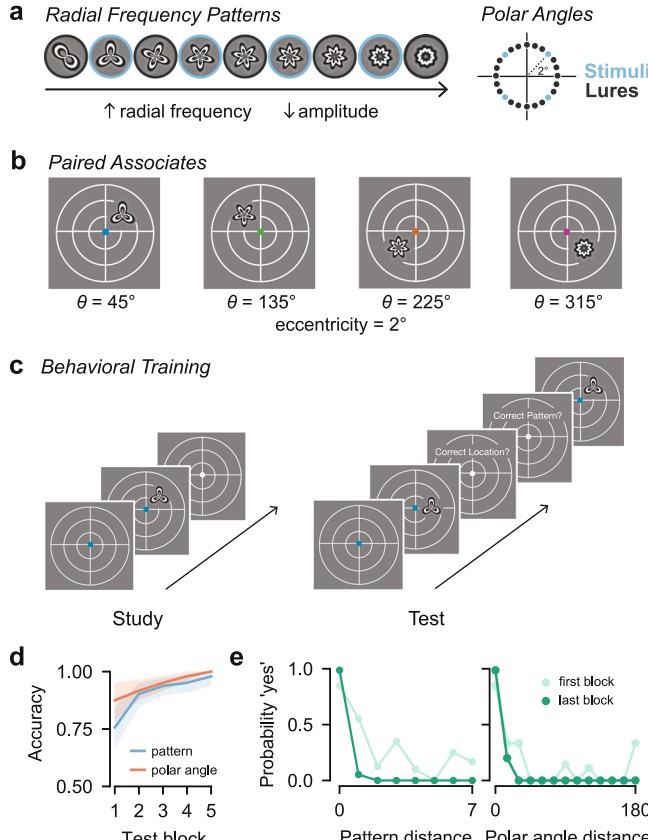

**Fig. 1 | Stimuli and behavioral training. a** The four radial frequency patterns and polar angle locations used in the fMRI experiment are outlined in blue. The intervening patterns and locations were used as lures during the behavioral training session. **b** In a behavioral training session immediately prior to the scan, participants learned that each of the four colored fixation dot cues was associated with a unique radial frequency pattern that appeared at a unique location in the visual field. **c** During training, participants alternated between study and test blocks. During study blocks, participants were presented with the associations while maintaining central fixation. During test blocks, participants were presented with the cues followed by test probes while maintaining central fixation. On each test trial, participants gave yes/no responses to whether the test probe was presented at the target polar angle and whether it was the target pattern. **d** Accuracy of pattern and polar angle responses improved over the course of the training session. Lines indicate mean accuracy across participants. The shaded region indicates 95% confidence interval. **e** Memory performance became more precise from the first to the last test block. During the first block, false alarms were high for stimuli similar to the target. These instances decreased by the last test block. Dots indicate probability of a 'yes' response for all trials in that bin in either the first or last block. The $x$ axis is organized such that zero corresponds to the target (yes = correct) and increasing values correspond to lures more dissimilar to the target (yes = incorrect).

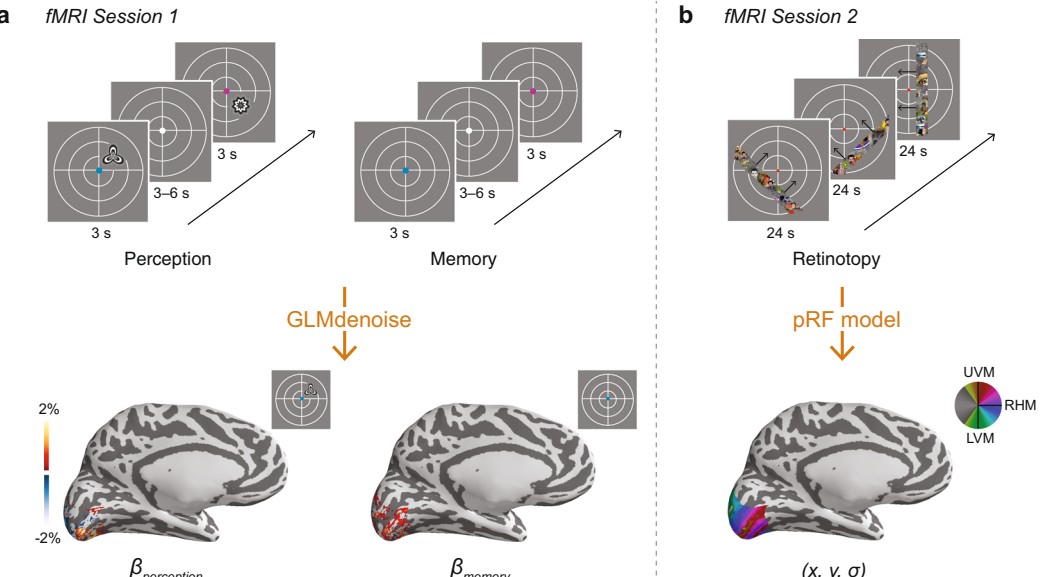

**Fig. 2 | fMRI task design and measurements. a** Following training, participants participated in two tasks while being scanned. During perception runs, participants viewed the colored fixation dot cues and associated stimuli while maintaining central fixation. Participants performed a one-back task on the stimuli to encourage covert attention to each stimulus. During memory runs, participants viewed only the cues and recalled the associated stimuli while maintaining central fixation. Participants made a judgment about the vividness of their memory (vivid, weak, no memory) on each trial. We used the perception and memory fMRI time series to perform a GLM analysis that estimated the response evoked by perceiving and remembering each stimulus for each vertex on the cortical surface. Responses in visual cortex for an example participant and stimulus are shown at bottom. **b** In a separate fMRI session on a different day, participants participated in a retinotopic mapping session. During retinotopy runs, participants viewed bar apertures embedded with faces, scenes, and objects drifting across the visual field while they maintained central fixation. Participants performed a color change detection task on the fixation dot. We used the retinotopy fMRI time series to solve a pRF model that estimated the receptive field parameters for each vertex on the cortical surface. A polar angle map is plotted for an example participant at bottom.

During fMRI memory runs, participants fixated on the central fixation dot cues and recalled the associated stimuli in their spatial locations. On each trial, participants made a judgment about the vividness of their memory. Participants reported that they experienced vivid memory on an average of 89.8% of trials (range: 72.4%–99.5%), weak memory on 8.9% of trials (0.5%–25.0%), and no memory on 0.5% of trials (0.0%–2.6%). Participants failed to respond on an average of 0.75% trials (0.0%–3.6%). We intentionally avoided asking participants to report the content of their memories while being scanned. We did this to avoid confounding the properties of the remembered stimulus with the motor response used to report these properties (hand or eye movement).

## Memory reactivation is spatially organized

We used a GLM to estimate the BOLD response evoked by seeing and remembering each of the four spatially localized stimuli (Fig. 2a; see Methods). Separately, each participant completed a retinotopic mapping session. We fit pRF models to these data to estimate pRF locations $(x, y)$ and sizes $(\sigma)$ in multiple visual areas (Fig. 2b). To more easily compare perception- and memory-evoked activity across visual areas, we transformed the evoked responses from cortical surface coordinates into visual field coordinates using the pRF parameters. For each participant, ROI, and stimulus, we plotted the amplitude of the evoked response at the visual field position $(x, y)$ estimated by the pRF model (Fig. 3a). We then interpolated these values over 2D space, rotated all stimulus responses to the upper visual meridian, z-scored the values, and averaged across stimuli and participants (see Methods). These plots are useful for comparison across regions because they show the organization of the BOLD response in a common space that is undistorted by the size and magnification differences present in cortex.

We generated these visual field plots for V1, V2, and V3 as an initial way to visualize the evoked responses during perception and memory. Readily apparent is the fact that stimulus-evoked responses during perception were robust and spatially specific (Fig. 3b, top). The spatial spread of perceptual responses increased from V1 to V3, consistent with estimates of increasing receptive field size in these regions[30,31]. While the memory responses were weaker and more diffuse, they were also spatially organized, with peak activity in the same location as the perception responses (Fig. 3b, bottom).

Next, we quantified these initial observations. Because our stimulus locations were isoeccentric, we collapsed our evoked responses to one spatial dimension: polar angle. To do this, we restricted our ROIs to surface vertices with pRF locations near the stimulus eccentricity, and then binned the responses according to their polar angle distance from the stimuli. We averaged the responses in each bin within a participant and then across participants. This produced polar angle response functions that mapped evoked BOLD response as a function of polar angle distance from the stimulus (see Methods; Fig. 4a). We generated these polar angle response functions for V1–V3 and for three mid-level visual areas from the ventral, lateral, and dorsal streams: hV4, LO, and V3ab (Fig. 4b). To capture the pattern of positive and negative BOLD responses we observed, we fit the average data in each ROI with a difference of two von Mises distributions, where both the positive and the negative von Mises were centered at the same location (see Methods). Visualizing the data and the von Mises fits (Fig. 4b), it's clear that both perception and memory fits show a peak at 0°, or the true location of the stimulus, in every region.

To formally test this, we calculated bootstrapped confidence intervals for the location parameter of the von Mises distributions by resampling participants with replacement (see Methods). We then compared the accuracy and reliability of location parameters between perception and memory (Fig. 4c, left). As expected, location parameters derived from perception data were highly accurate. 95% confidence intervals for perception location parameters overlapped 0° of polar angle, or the true stimulus location, in all ROIs. These confidence intervals spanned only 7.0° on average (range: 3.9°–9.5°), demonstrating that there was low variability in location accuracy across

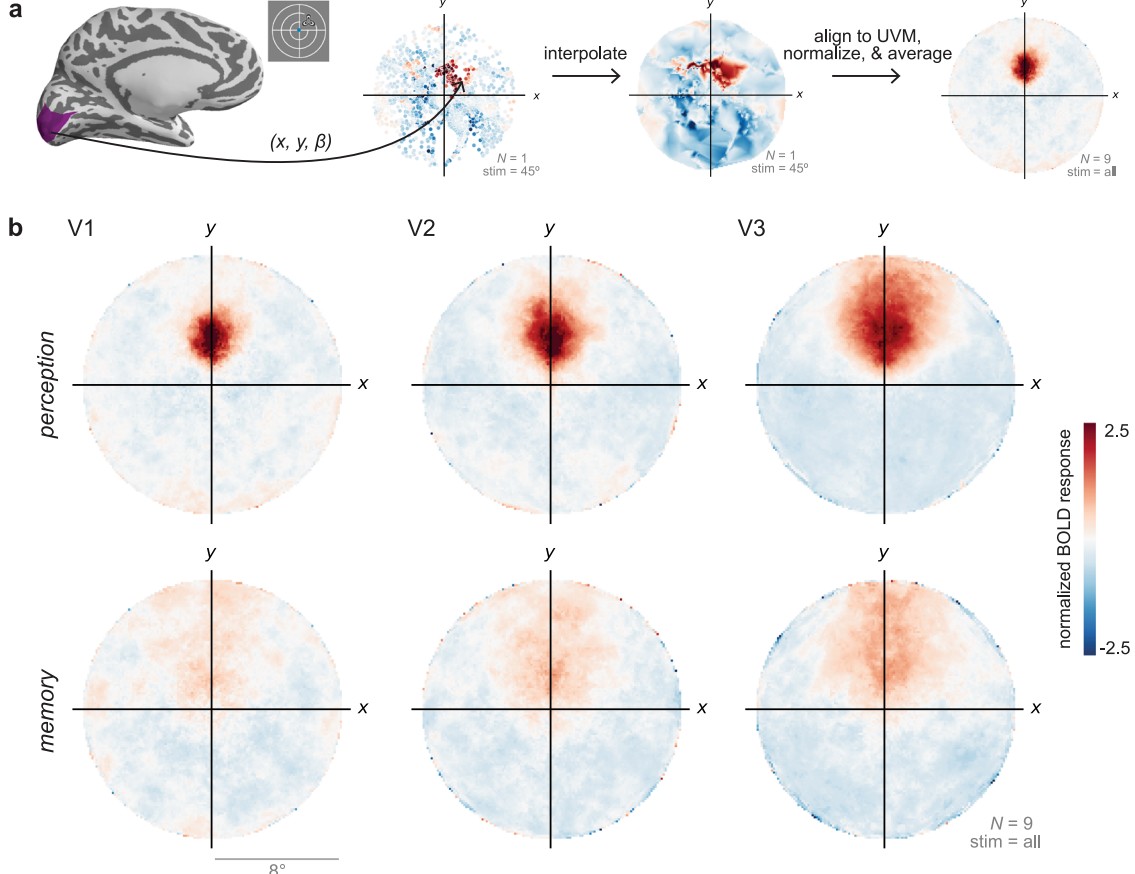

**Fig. 3 | Perception and memory activity in visual field coordinates. a** For every participant, ROI, and stimulus, we plotted the perception- or memory-evoked response ($\beta$) in the visual field position estimated by the pRF model ($x, y$) and then interpolated over 2D space. Hotter colors indicate larger BOLD responses in that part of the visual field. We then rotated the interpolated images by the polar angle location of the stimulus so that they aligned on the upper vertical meridian. Finally, we *z*-scored the images and then averaged over stimuli and participants to produce on average image per ROI and task. An example participant's V1 data during the perception task is shown in the two middle panels, and the group average V1 data during perception is shown at right. **b** Plots of perception-evoked and memory-evoked activity, averaged across all participants ($N = 9$), from V1, V2, and V3. These plots reproduce known features of spatial processing during perception, such as increasing receptive field size from V1--V3. They also qualitatively demonstrate that perceptual activity is not perfectly reproduced during memory retrieval but that some retinotopic organization is preserved.

participants in every ROI. Critically, memory parameters were also highly accurate, with confidence intervals overlapping 0° in every ROI (Fig. 4c, left). Thus, in every visual area measured, the spatial locations of the remembered stimuli could be accurately estimated from mnemonic activity. Memory confidence intervals spanned 17.6° on average (range = 11.0°–21.3°), indicating that location estimates were somewhat less reliable during memory during perception. However, even the widest memory confidence interval spanned far less than the 90° separating each stimulus location, demonstrating that there was no confusability between stimuli present in distributed memory activity. Because both perception and memory location parameters were highly accurate, and because differences in reliability were relatively small, there was no statistically reliable difference between perception and memory in the estimated location of peak activity (main effect of perception/memory: $\beta = 0.14$, 95% CI = [−6.72, 6.14]; Fig. 4c, left). Similarly, there was no reliable difference across ROIs (main effect of ROI: $\beta = 0.075$, 95% CI = [−2.22, 2.65]) and no reliable change in the relationship between perception and memory across ROIs (perception/memory x ROI interaction: $\beta = 0.54$, 95% CI = [−1.94, 2.60]). These results provide evidence that memory-triggered activity in human visual cortex is spatially organized within known visual field maps, as it is during visual perception. These findings support prior reports of retinotopic activity during memory and imagery[5,9,15], but provide more quantitative estimates of this effect.

## Amplitude and precision differ between perceptual and mnemonic activity

Aspects of perception and memory responses other than the peak location differed considerably. First, the strongest memory responses were lower in amplitude than the strongest perception responses (Fig. 4b). To quantify this observation, we derived a population measure of response amplitude (maximum response−minimum response) from the difference of von Mises functions fit to our data (see Methods). This measure indicated the relative strength of the BOLD response in vertices that were maximally responsive to stimulus. As described in the prior section, these vertices almost always corresponded to the polar angle location of the stimulus (0° of polar angle distance). We computed bootstrapped confidence intervals for this amplitude measure, following the prior analysis. We then compared these estimates between perception and memory. First, response amplitudes for perception data were higher than for the memory data (main effect of perception/memory: $\beta = 0.95$, 95% CI = [0.80, 1.13]; Fig. 4c, middle). The average amplitude during perception was 0.92 units of BOLD % signal change, and the average amplitude during memory was 0.26 units of BOLD % signal change. Amplitude confidence intervals for perception and memory did not overlap in any ROI, indicating that these differences were highly reliable in each region. Critically, the fact that perception amplitudes were larger than memory amplitudes overall does not imply that memory responses

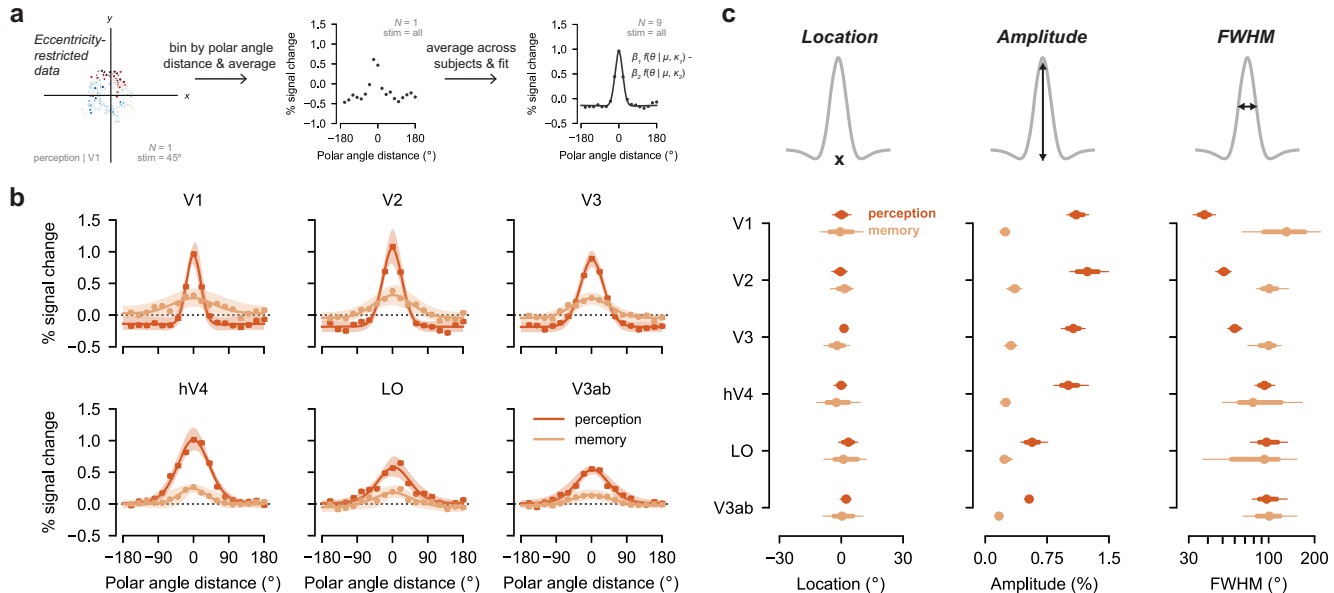

**Fig. 4 | Perception and memory have shared and distinct activation features in visual cortex. a** We created 1D polar angle response functions by restricting data to vertices with eccentricities near the stimulus and then binning the vertices according to polar angle distance from the stimulus. We computed the median evoked BOLD % signal change within these bins for each participant, ROI, task. We then performed a normed mean across participants. A difference of two von Mises distributions was fit to the group average response. The von Mises distributions over θ = [−180°, 180°] shared a location parameter (μ) but could differ in their concentration (κ₁, κ₂) and scale (β₁, β₂). An example participant's V1 data during the perception task is shown in the two left panels, and the group average V1 data during perception is shown at right with fit von Mises. **b** Group polar angle response functions are plotted separately for perception and memory. Dots represent group average BOLD % signal change at different polar angle distances from the stimulus. Responses in parts of cortex that have pRFs near the stimulus position are plotted

at x = 0. Lines represent the fit of the difference of two von Mises distributions to the average data, and shading represents the 95% confidence interval around this fit. While the peak location of the response is shared across perception and memory, there are clear differences in the amplitude and width of the responses. **c** Location, amplitude and FWHM of the difference of von Mises fits to the group data (N = 9) are plotted to quantify the responses. Dots represent fit parameters and lines represent bootstrapped 68% confidence intervals (thick lines) and boot-strapped 95% confidence intervals (thin lines) generated from resampling partici-pants with replacement. In all ROIs, the peak location of the response is equivalent during perception and memory (at 0°, the stimulus location), while the amplitude of the response is reliably lower during memory than during perception. The FWHM of the response increases across ROIs during perception but not during memory, resulting in highly divergent FWHM for perception and memory in early visual areas.

were at baseline. In fact, 95% confidence intervals for memory ampli-tudes did not overlap with zero in any region (Fig. 4c, middle), demonstrating that memory responses were above baseline in all areas measured. These results demonstrate that the amplitude of spatially organized activity in visual cortex is attenuated, but present, during memory retrieval.

During both perception and memory, we observed lower response amplitudes in later areas (main effect of ROI: β = −0.023, 95% CI = [−0.035, −0.008]; Fig. 4c, middle). However, there was a larger decrease in amplitude from early to late areas during perception than during memory (perception/memory x ROI interaction: β = −0.12, 95% CI = [−0.16 −0.082]). This resulted in perception and memory ampli-tudes that were more similar to each other in later visual areas than in early visual areas. For example, while perception amplitudes were 0.86 units of % signal change larger than memory amplitudes in V1, they were only 0.37 units of % signal change larger in V3ab. This pattern of data indicates that the extent to which memory amplitudes recapitu-late perception amplitudes varies across the visual system.

Memory responses were also wider, or less precise, than percep-tion responses (Fig. 4b). We operationalized the precision of percep-tion and memory responses by computing the full width at half maximum (FWHM) of the difference of von Mises fit to our data and by generating confidence intervals for this measure. Note that FWHM is not sensitive to the overall scale of the response function: a difference of von Mises rescaled by a factor of 0.5 will have an unchanged FWHM. On average, FWHM during perception was significantly smaller than during memory (main effect of perception/memory: β = −75.2, 95% CI = [−138.5, −33.1]; Fig. 4c, right). However, these differences were not

equivalent across ROIs (perception/memory x ROI interaction: β = 18.8, 95% CI = [5.78, 35.5]). During perception, FWHM increased moving up the visual hierarchy (main effect of ROI: β = 13.3, 95% CI = [10.3, 20.6]), indicating increased width or decreased precision in later visual areas compared to early visual areas (Fig. 4c, right). For example, V1 had the narrowest (most precise) response during perception, with an average FWHM of 38.0° (95% CI: [32.0°, 45.0°]), while V3ab had the widest responses during perception, with a FWHM of 97.0° (95% CI: [78.0°, 132.5°]). This increasing pattern follows previously described increases in population receptive field size in these regions[30,31]. Note that a separate question, not addressed here, is the precision with which the stimulus can be decoded from a representation, which is not neces-sarily related to receptive field size.

Strikingly, however, this pattern of increasing FWHM from early to late visual areas was abolished during memory retrieval (main effect of ROI: β = −5.49, 95% CI = [−18.7, 8.41]; Fig. 4c, right). In fact, numerically, V1 had the largest FWHM of all ROIs during memory retrieval at 131.0° (95% CI: [66.9°, 225.0°]), a reversal of its rank during perception, with V2, V3, and hV4 FWHMs getting increasingly smaller. Though decreasing FWHM from early to late areas was not statistically reliable in this sample, the clear absence of a perception-like pattern during memory retrieval is notable. This observation suggests that funda-mental aspects of spatial processing commonly observed during per-ception do not generalize to memory-evoked responses.

Similar to our findings for amplitude, the interaction between perception/memory and ROI yielded the most divergent perception and memory responses in the earliest visual areas (Fig. 4c, right). V1 responses during memory were 3.45 times wider than V1 responses

during perception. In V2 and V3, memory FWHM exceeded perception FWHM by an average of 1.98 times and 1.67 times, respectively. In hV4, LO, and V3ab, memory responses were 0.84–1.04 times wider during memory than during perception. This increase in similarity between perceptual and mnemonic responses in later areas raises the interesting possibility that later stages of visual processing serve as a bottleneck on mnemonic activity precision. Finally, we note that changes in FWHM can also be understood as distance-dependent changes in the strength of the BOLD response. Consider the change between perception FWHM and memory FWHM in V1. Among vertices with pRF centers that were near the stimulus (polar angle distance = 0˚), BOLD responses were larger during perception than memory, as described in the previous section on amplitude. However, for voxels farther away from the stimulus, BOLD responses were actually larger during memory than perception (Fig. 4b). This pattern argues against the notion that memory-evoked responses are merely low amplitude copies of perceptual responses. Instead, our results provide evidence for reliable and striking differences in the precision of perception and memory activity across different levels of the visual system. These findings suggest that there are fundamentally different constraints on the properties of feedforward perceptual activity and top-down mnemonic activity in human visual cortex.

We've focused on polar angle responses for several reasons. First, our stimuli varied along this dimension. Second, polar angle estimates are more reliable than eccentricity estimates in human visual cortex, especially in hV4, LO, and V3ab[36]. Nonetheless, we also plotted eccentricity responses and observed qualitatively similar patterns to those in the polar angle domain (Supplementary Fig. 1). During perception, eccentricity tuning became increasingly wide in later areas. During memory, eccentricity tuning was much wider overall, with the biggest differences between perception and memory in the earliest areas. Similar to our polar angle analyses, V1 vertices with pRFs far away from the stimulus eccentricity actually had numerically higher responses during memory than during perception. Though these data are noisier than our polar angle data and were not the original target of our investigation (stimulus eccentricity was not experimentally manipulated), the results provide some preliminary evidence that our polar angle results generalize to other stimulus properties.

### Differences between perception and memory responses are not explained by inter-participant variability
The properties of group data are not always reflected in individuals, who sometimes deviate qualitatively from the average[37]. Could our observation of different amplitude and FWHM during perception and memory be explained by greater inter-participant variability during the memory task than during the perception task? For example, is it possible that memory responses were as precise as perception responses when considered in individual participants, but that the group average memory response was wide due to different peak locations in different participants? To answer this question, we examined individual participant data to confirm that the findings present in the group average data were present in individuals. Instead of averaging the BOLD responses across participants prior to fitting the difference of von Mises functions, we fit each participant separately after removing baseline offsets (Supplementary Fig. 2a, b; see Methods). This yielded location, amplitude, and FWHM measures (as described in the prior section) for each participant, ROI, and task (Supplementary Fig. 2c). When we examine individual participant estimates, we see good agreement with our group average results. In V1, 7/9 individual participants had larger FWHM during memory than during perception. Averaging across all nine individual participants, V1 FWHM was larger during memory than during perception by a factor of 2.5 times. Similarly, in V2 and V3, 9/9 and 7/9 participants had larger FWHM during memory than perception, with an average factor of 2.14 times and 1.51 times, respectively. In contrast, in hV4, LO, and V3ab, 5/9

participants had larger FWHM values during memory than perception, with an average factor of between 1.02 and 1.20 times. These findings corroborate our analysis of the group fits in the prior section (Fig. 4).

To further validate our group-level findings, we ran repeated measures ANOVAs on individual participant parameters. We confirmed that FWHM was larger during memory than during perception (main effect of perception/memory: $F_{1,8} = 16.8$, $p = 0.003$) and that this effect interacted with ROI such that the biggest difference between perception and memory FWHM was in early areas (perception/memory x ROI interaction: $F_{1,8} = 33.3$, $p < 0.001$). Likewise, we confirmed that amplitudes were larger during perception than during memory (main effect of perception/memory: $F_{1,8} = 175.5$, $p < 0.001$), and that this effect also interacted with ROI in the same way (perception/memory x ROI interaction: $F_{1,8} = 66.5$, $p < 0.001$). There was no reliable difference in peak locations during perception and memory (main effect of perception/memory: $F_{1,8} = 0.053$, $p = 0.820$). All of these effects were preserved when ROI was coded categorically rather than numerically according to visual hierarchy position and when correcting for marginal violations of sphericity (see Methods). In summary, we find no evidence that inter-participant variability can explain our results. The effects we report in the prior section can be readily observed at the individual participant level.

### Differences between perception and memory responses are not explained by trial-to-trial noise
Another consideration in interpreting our results is whether the differences we observed between perception and memory could be caused by differences in trial-to-trial noise between the two tasks. For example, is it possible that perception and memory responses were actually equivalent other than noise level, but that greater trial-to-trial noise caused memory responses to appear to have systematically different tuning? In particular, we sought to understand whether differences in memory responses could be explained by two types of noise: measurement noise, caused by low amplitude fMRI signals, and memory noise, caused by task failures during the memory task. To this end, we simulated the effect of four different noise sources on our data: (1) reduced fMRI signal-to-noise (measurement noise); (2) retrieval task lapses (memory noise); (3) associative memory errors (memory noise); (4) angular memory errors (memory noise). If perception and memory have the same fundamental response properties, but the memory task is subject to more noise, then adding noise to the perception data should yield responses that look like what we observed during memory.

To investigate this possibility, we started with perception data (mean and variance of each vertex's activity during perception) and tested whether we could generate responses that looked like memory data by adding noise from one of the four noise sources. To simulate reduced fMRI signal-to-noise, we introduced additive noise to each vertex's perception response (Fig. 5a, left; see Methods). To simulate retrieval task lapses, we created some trials where the mean response was zero (Fig. 5b, left). To simulate associative memory errors, we replaced some perception responses with responses corresponding to one of the other studied stimuli (Fig. 5c, left). To simulate angular memory error, we added angular noise to the peak location of the perception responses (Fig. 5d, left). For each of these types of simulation, we considered multiple levels of noise. To be conservative, we simulated responses that were highly correlated across a region of interest (see Methods), which ensured that vertex-level noise would carry forward to our population measures. To assess our simulation results, we analyzed all simulated datasets with the same procedures used for the real data and then plotted the resulting von Mises fits (Supplementary Fig. 3) and parameters (Supplementary Fig. 4). We then counted the proportion of times the von Mises parameters derived from a simulation fell within the 95% confidence interval of the actual memory data (Fig. 5a–d, right).

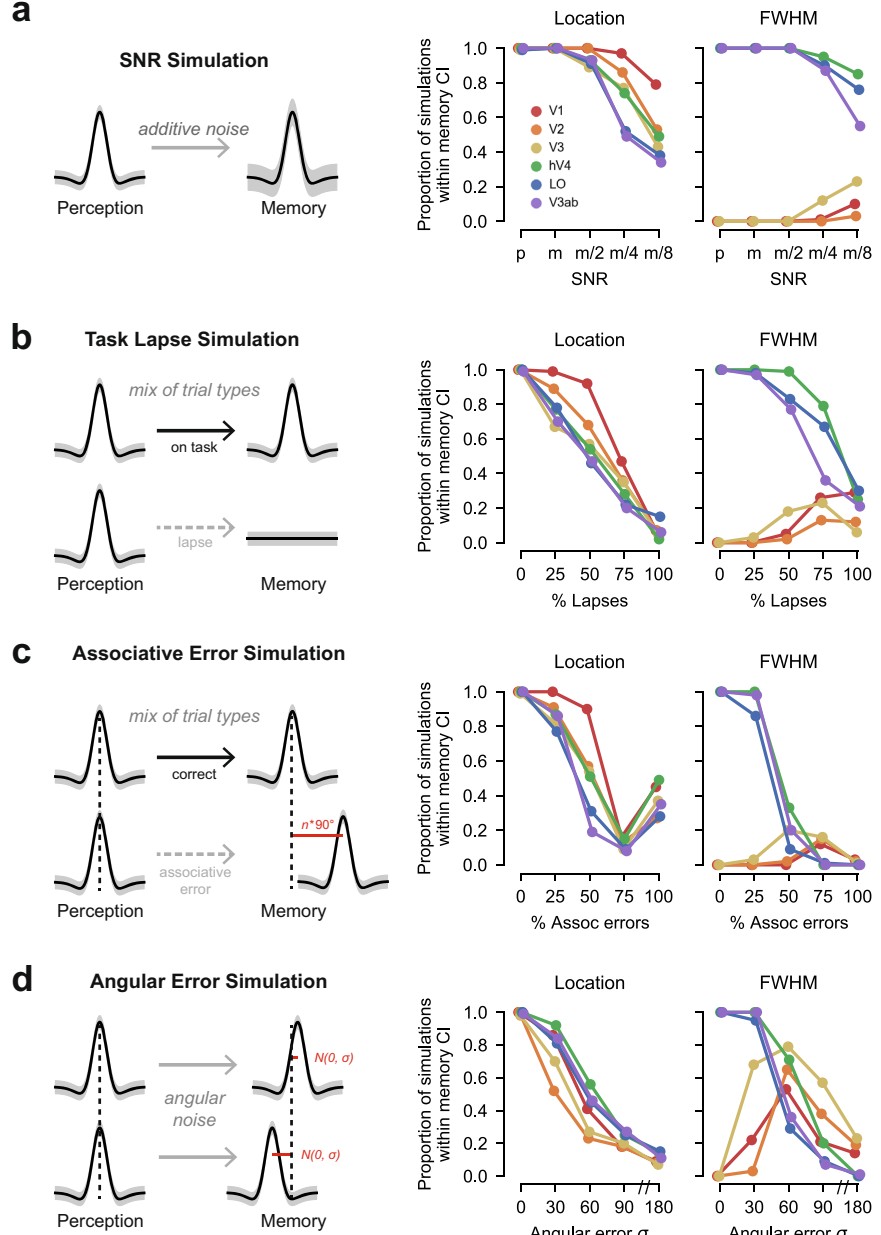

**Fig. 5 | Differences between perception and memory are not explained by noise. a** Left: We simulated the effect of low SNR by introducing additive noise to our perception data and testing whether this produced responses similar to what we observed during memory. Right: The proportion of simulations that produce location and FWHM parameters within the 95% confidence intervals of the memory data are plotted for decreasing signal-to-noise ratios (SNR) and for each ROI. SNR values include the empirical SNR of the perception data (p), the empirical SNR of the memory data (m), and 1/2, 1/4, and 1/8 of the empirical SNR of the memory data. **b** Left: We simulated the effect of retrieval task lapses by generating perception datasets where a subset of trials had a mean BOLD response of zero. Right: Data are plotted as in a, with increasingly large numbers of lapsed trials on the *x* axis. **c** Left: We simulated the effect of associative memory errors by substituting some perception responses with responses that corresponded to one of the other three

studied stimuli. Errors of this type were always a multiple of 90°, which was the distance between studied stimuli. Right: Data are plotted as in a, with increasingly large numbers of associative errors on the *x* axis. **d** Left: We simulated the effect of angular memory errors by adding angular noise to the peak location of the perception responses. Angular memory errors were produced by drawing a new peak location value from a normal distribution centered at the true stimulus location and with some width, *σ*. Right: Data are plotted as in a, with increasing large *σ* values on the x axis. In all panels, high noise levels are required to generate FWHM parameters within the confidence intervals of the memory data in V1-V3. In addition, in all panels, high noise levels produced poor matches to memory FWHM in hV4, LO and V3ab, as well as unreliable location parameters in all ROIs. See Supplementary Fig. 3 for visualization of fitted von Mises for each simulation and Supplementary Fig. 4 for the full distribution of FWHM estimates for each simulation.

We began with the SNR simulation. First, using bootstrapped parameter estimates, we confirmed that the estimated signal-to-noise ratio (see Methods) for perception parameter estimates was higher than for memory parameter estimates in every ROI. Perception SNR was between 1.2 and 1.6 times higher than memory SNR in each ROI, and between 2.2 and 4.3 times higher in vertices closest to the stimulus location. To evaluate the impact of this difference on our results, we

simulated new perception data that precisely matched the empirical SNR of our memory data for every surface vertex. We also simulated data with even lower SNR (higher noise) than what we observed during memory. As expected, simulating perception data with reduced SNR increased variance in the location, amplitude, and FWHM of the von Mises fits (Supplementary Figs. 3a and 4a). However, no level of SNR produced response profiles that matched the memory data well. In

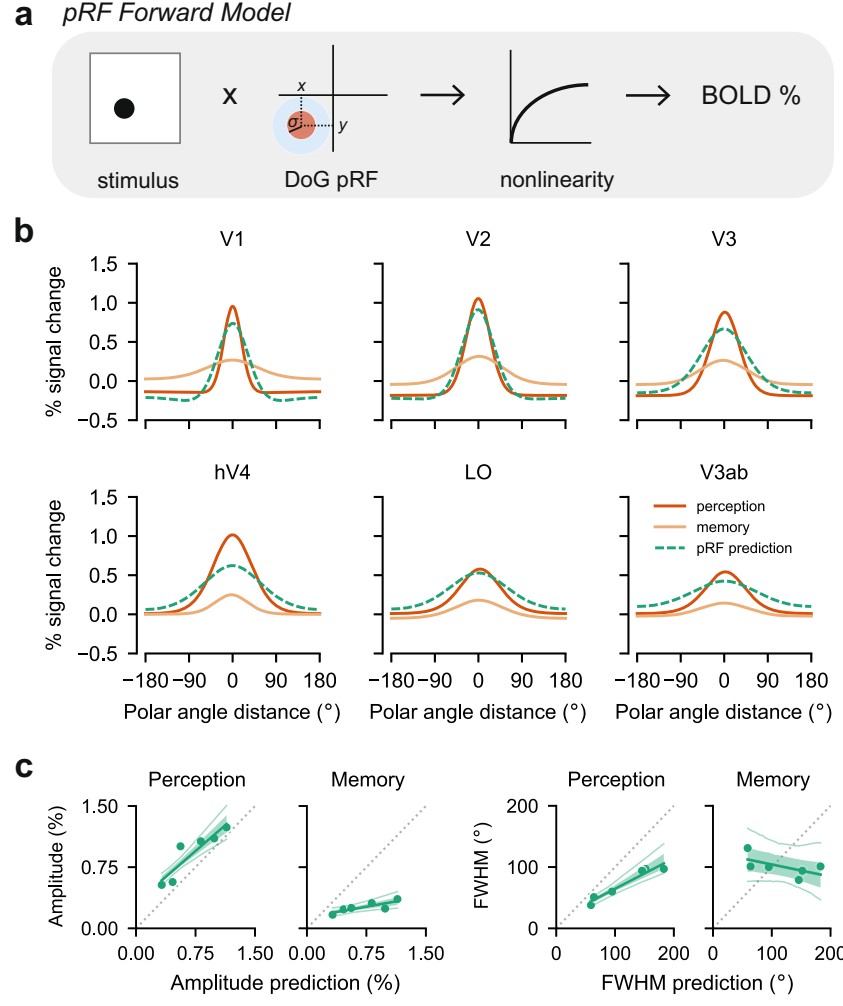

**Fig. 6 | pRF forward model captures perception but not memory responses. a** We used our pRF model to generate the predicted BOLD response to each of our experimental stimuli. The model assumes a Difference of Gaussians pRF shape, with a fixed positive to negative Gaussian size ratio (1:2) and amplitude ratio (2:1). The model also incorporates a compressive nonlinearity. **b** Polar angle response functions predicted by the pRF model (green dashed lines) are plotted alongside the response functions generated from the data (dark and light orange, reproduced from Fig. 4b). The model predictions are closer to the perception data than the memory data in all visual areas. **c** Predicted versus observed amplitude (left) and FWHM (right), plotted separately for perception and memory. Each dot represents an ROI. Lines represent the line of best fit across the dots. The shaded region is the 68% confidence interval generated from bootstrapping across participants, and the thin lines indicate the 95% confidence intervals. Perfect predictions will lie on the dashed gray lines. For both the amplitude and FWHM, the perception data lie relatively close to the pRF model predictions, whereas the memory data do not.

V1—the region where we observed the largest difference in FWHM between perception and memory—0% of the FWHM parameters in the memory SNR simulation approximated the actual memory data (Fig. 5a, right). In the noisiest simulation we performed (1/8 of the memory SNR), this figure was still only 10% (Fig. 5a, right). Similar results occurred for V2 and V3. These simulations demonstrate that low SNR cannot explain the pattern of memory responses we observed in early visual cortex.

Our SNR simulations also demonstrate that there are fundamental tradeoffs between capturing memory FWHM in early visual cortex and in capturing other aspects of the data. First, at high levels of noise (low levels of SNR), any modest increase in ability to capture V1-V3 FWHM was accompanied by a decrease in ability to capture FWHM in later visual areas (Fig. 5a, right, and Supplementary Figs. 3a and 4a). In these later regions, FWHM was empirically equivalent during perception and memory, and artificially adding noise to the perception data eliminates this equivalence. Second, high noise simulations generated more variability in the location parameters than was actually observed in the memory data in all ROIs (Fig. 5a, right).

We observed a similar pattern of results in the retrieval task lapse simulations. Very high lapse rates were required to generate any FWHM parameters that were sufficiently wide to match the memory data in V1 (Supplementary Figs. 3b and 4b). Only when simulating lapses in 50% of all trials, did this number exceed 0% (Fig. 5b, right). This frequency of task lapse is out of line with participants' actual lapse rate of 1.3% (self-reported no memory trials plus no response trials). Similar to the SNR simulation, any improvement in ability to capture the V1 FWHM data with increased lapses was offset by a decline in ability to capture FWHM in late visual areas (Fig. 5b, right), where responses became much wider than what was observed empirically during memory. Again, as in the SNR simulation, high rates of retrieval task lapse were associated with location parameters that were far noisier than what we observed during memory (Fig. 5b, right). These simulations demonstrate that retrieval task lapses are unlikely to explain the pattern of memory responses we observed in early visual cortex.

Next, we considered the associative error simulation. Very high associative error rates were required to generate any FWHM parameters that were sufficiently wide to match the memory data in V1

(Supplementary Fig. 3c and 4c). Only when simulating associative errors in >50% of all trials, did this number exceed 0% (Fig. 5c, right). Participants made zero associative memory errors (Fig. 1e) in the final round of training prior to the scan, making this level of poor performance unlikely. Further, similar to the prior simulations, a large number of associative errors decreased our ability to approximate memory FWHM in late areas and generate accurate memory location estimates (Fig. 5c, right). These simulations show that associative memory errors are unlikely to explain our data.

Finally, we considered the angular memory error simulation. Compared to the other simulations, this simulation produced a better match to memory FWHM in V1 when assuming high levels of noise (Supplementary Figs. 3d and 4d). In the best performing simulation, 53% of the V1 FWHM parameters approximated the memory data (Fig. 5d, right). However, the magnitude of memory error in this simulation was implausibly high. The standard deviation of memory errors around the true value was 60°, meaning that simulated memories were within the correct quadrant <60% of time. Given that participants were trained to discriminate remembered locations up to 15° (see Methods), errors of this magnitude and frequency are exceedingly unlikely. Similar to the previous simulations, improvements in the ability to capture V1 memory FWHM with high levels of angular error were associated with large decreases in the ability to capture other aspects of the memory data (Fig. 5d, right). Like previous simulations, these simulations show that angular memory error is an unlikely explanation for our observed results.

Collectively, these simulations demonstrate that our results are unlikely to be caused by reduced SNR or various forms of memory failure. In each of the four simulations, the amount of noise required to make even modest gains in our ability to account for the V1 memory FWHM was implausibly large. Further, in all four cases, increases in the ability to account for V1 FWHM were accompanied by decreases in the ability to account for FWHM in higher visual areas and to recover location parameters that were as reliable as our actual data.

## pRF models accurately predict perception but not memory responses

Next, we evaluated how well perception and memory responses matched the predictions of a pRF model. To do this, we used each participant's pRF model to generate predicted cortical responses to each of the four experimental stimuli (Fig. 6a). The pRF model we used to generate predictions had a compressive nonlinearity[31] and difference of Gaussians (DoG) pRF shape[38] with a fixed positive to negative Gaussian ratio (see Methods). The predictions from the model were analyzed with the same procedure as the data, yielding von Mises fits to the predicted data (Fig. 6b). Model predictions from simpler pRF models are shown in Supplementary Fig. 5a.

Qualitatively, the pRF model predictions agree with the perception data but not the memory data (Fig. 6b). Several specific features of the perception data are well captured by the model. First, the model predicts the highest amplitude response at cortical sites with pRFs near the stimulus location (peak at 0°). Second, the model predicts increasingly wide response profiles from the early to late visual areas. Third, it predicts higher amplitudes in early compared to late areas. Finally, the model predicts negative responses in the surround locations of V1–V3 but not higher visual areas. This is notable given that all voxel pRFs were implemented with a negative surround of the same size and amplitude relative to the center Gaussian. This suggests that voxel-level parameters and population-level responses can diverge[39]. Though not the focus of this analysis, we note that the model predictions are not perfect. The model predicts slightly lower amplitudes and larger FWHM than is observed in the perception data (Fig. 6b). These discrepancies may be due to differences between the stimuli used in the main experiment and those used in the pRF experiment,

differences in the task, or the fixed ratio between the positive and negative Gaussians in the pRF.

Critically, the model accurately captures the properties of memory responses that are shared with perception responses (the peak location), but not the distinct properties (amplitude and FWHM; Fig. 6b). These failures are especially clear when comparing the predicted amplitude and FWHM from the pRF model with the observed amplitudes and FWHMs for perception and memory. While there is a positive slope between the predicted amplitudes and both the perception amplitudes ($\beta = 0.84$, 95% CI: [0.56, 1.15]) and memory amplitudes ($\beta = 0.17$, 95% CI: [0.056, 0.32]), the slopes differ substantially (Fig. 6c). The perception amplitudes have a slope closer to 1, indicating good agreement with the model predictions, while the memory data have a slope closer to 0, indicating weak agreement. Similarly, the predicted FWHM is significantly and positively related to the perception FWHM ($\beta = 0.50$, 95% CI: [0.36, 0.76]), but weakly and negatively related to the memory FWHM ($\beta = -0.20$, 95% CI: [−0.67, 0.26]; Fig. 6c). Thus, the pRF model predicts the pattern of increasing amplitude and FWHM that we observed across ROIs during perception but does not predict the patterns we observed during memory retrieval. In order to rule out the possibility that this finding is specific to the particular pRF model we used, we performed the same analysis on two simpler, previously published, pRF models (a CSS model and a linear model; see Methods). We find that these models are also dramatically better at capturing across-ROI patterns in FWHM and amplitude for the perception data than for the memory data (Supplementary Fig. 5b).

In addition to assessing how well our pRF model captured across-ROI changes in FWHM and amplitude, we directly quantified the goodness-of-fit between the pRF model predictions and our experimental data in each ROI. To do this, we computed the $R^2$ between the predicted and observed polar angle response functions, separately for each ROI and task. Goodness-of-fit was very high for the perception data in every ROI (range: 0.65–0.93). In contrast, goodness-of-fit was poor for the memory data. $R^2$ values were negative in every ROI during memory (range: −11.9 to −1.84), indicating that the pRF model failed to capture the mean response (Supplementary Fig. 6, top). This is largely driven by the fact that the mean predicted amplitude from the pRF model was much higher than the mean amplitude of the memory data. We wondered to what extent this failure could be corrected by rescaling the pRF predictions to be lower in amplitude. After rescaling the pRF predictions with the single scale factor that best approximated the perception data or the memory data, we re-evaluated $R^2$. While rescaling improved $R^2$ for the memory data in every ROI, $R^2$ for the memory data still fell below $R^2$ for the perception data in most ROIs, with the largest failure in V1 (Supplementary Fig. 6, bottom). These analyses demonstrate that our pRF model is a good predictor of the perceptual responses we measured, but a poor predictor of the mnemonic responses we measured.

Together, these results support our interpretation of the data in Fig. 4 to mean that memory and perception have distinct spatial tuning properties. The critical advantage of using pRF models is that they explicitly incorporate known properties of feedforward spatial processing in visual cortex. Because our pRF model fails to account for the memory responses we observed, we can conclude that memory reactivation violates the assumptions of feedforward processes that accurately characterize perceptual activation. A plausible explanation for this failure is that memory retrieval involves a fundamentally different origin and cascade of information through visual cortex, a possibility we explore in the next section.

## Perception and memory responses can be simulated with a bidirectional hierarchical model

Cortical activity during perception arises from a primarily feedforward process that originates with the retina and that accumulates additional spatial pooling in each cortical area, resulting in increasingly large

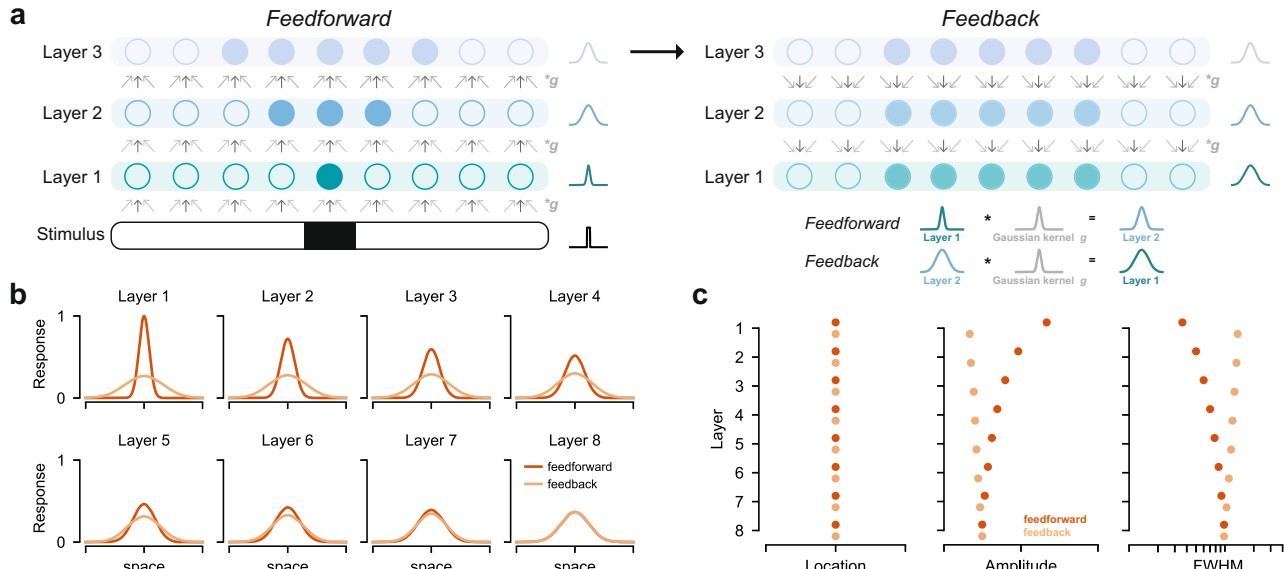

**Fig. 7 | Perception and memory responses can be simulated with a bidirectional hierarchical model. a** Illustration of stimulus-driven activity propagating through a linear hierarchical network model in the feedforward direction (left) and mnemonic activity propagating through the model in the feedback direction (right). In both cases, a given layer's activity is generated by convolving the previously active layer's activity with a fixed Gaussian kernel. The feedforward simulation began with a boxcar stimulus. The feedback simulation began with duplication of the feedforward activity from the final layer. **b** Example of feedforward and feedback simulations for one set of parameters (stimulus = 15°; kernel $\sigma$ = 15°, number of layers = 8), plotted in the conventions of Fig. 4b. The feedforward simulation parallels

our observations during perception, and the feedback simulation parallels our observations during memory. **c** Location, amplitude, and FWHM parameters for each layer, plotted separately for feedforward and feedback simulations in **b**. Location is preserved across layers in the feedforward and feedback direction. Note that FWHM becomes progressively wider in later layers in the feedforward direction and in earlier layers in the feedback direction. This results in large differences in FWHM between feedforward and feedback activity in early layers. These trends closely follow our observations in Fig. 4c. See Supplementary Fig. 7 for other simulations with different layer numbers, stimulus sizes, and kernel sizes.

receive fields[30,40]. In contrast, memory reinstatement is hypothesized to depend on the hippocampus[41,42], a region bidirectionally connected to high-level visual areas in ventral temporal cortex via the medial temporal lobe cortex[43–45]. Reinstated cortical activity is then thought to propagate backwards through visual cortex[46–49], driven by the hippocampus and/or its cortical connections. Here, we explored whether a simple hierarchical model with spatial pooling could capture the qualitative pattern of results we observed. We were specifically interested in whether a single model could account for changing patterns of precision and amplitude between perception and memory retrieval. We asked whether manipulating the direction of information flow within the model would be sufficient to account for these qualitative differences without changing any components of the model structure or parameter values.

Importantly, we did not fit our model to our data. Nor did we attempt to generate a model that would predict the exact FWHM and amplitude values we report. Instead, we sought to identify a class of models where most parameter choices yielded the qualitative pattern of data we observed during both perception and memory. To do this, we first constructed the general form for a feedforward-only hierarchical model of spatial processing in neocortex. Given that feedforward convolutional networks are widespread in visual neuroscience and computer vision[50], we expected even a highly simplified model to produce spatial responses qualitatively similar to our experimental observations from the perception task. We constructed our model such that the activity in each layer was created by convolving the activity from the previous layer with a fixed Gaussian kernel (Fig. 7a, left; see Methods). Beginning with a boxcar stimulus, we cascaded this convolutional operation to simulate multiple layers of the network. Note that specific layers of the model do not have a one-to-one correspondence with any stage of visual processing or cortical area. We varied the size ($\sigma$) of the convolutional kernel, the size of the stimulus (width of boxcar), and

the number of layers in the model in order to explore the full parameter space of this model. We plot one instance of the model with 8 layers, a 15° stimulus, and kernel with $\sigma$ = 15° in Fig. 7b. As expected, the location of the simulated peak response during feedforward processing was unchanged across layers, but the FWHM of the responses increased (Fig. 7c). This qualitatively matches the pattern of fMRI responses we observed during our perception task (Fig. 4b, c). This increasing FWHM pattern during feedforward simulations holds true for a wide range of model parameters (Supplementary Fig. 7a, b).

We then explored whether backwards propagation of reinstated activity in our hierarchical model could account for patterns in our memory data. To do this, we assumed that feedforward and feedback connections in the model were reciprocal, meaning that the convolutional kernel was the same in feedforward and feedback direction. We assumed perfect reinstatement in the top layer, and thus began the feedback simulation by duplicating the feedforward activity from the final layer. Starting with this final layer activity, we convolved each layer's activity with the same Gaussian kernel to generate earlier layers' activity (Fig. 7a, right). The properties of the simulated activity (Fig. 7b, c) in our example model instance bear a striking resemblance to those of the observed memory data (Fig. 4b, c). First, simulated feedback activity had a preserved peak location across layers (Fig. 7c, left), similar to the memory data. Second, simulated feedback activity was wider and lower amplitude than feedforward activity overall (Fig. 7c, middle and right)—just as our memory data had wider and lower amplitude responses than our perception data. Third, the increase in FWHM across layers was smaller in the feedback direction than in the feedforward direction, and it reversed direction with respect to the visual hierarchy. This reversal is of interest given that this trend was numerically present in our memory data but small in magnitude and not statistically reliable at our sample size. Finally, the difference between feedforward and feedback amplitude and FWHM was

maximal in the earliest layers, just as the difference between our perception and memory data was maximal in V1.

We note that while the exact FWHM values depend on the number of layers, the stimulus size, and the kernel size (Supplementary Fig. 7), the qualitative patterns we focus on are general properties of this model class and can be observed across many parameter choices. Though some parameter choices can result in little to no change in FWHM, there are no parameter choices that can reverse the trends we report – for example, by producing smaller FWHM in layer 1 than in layer 2 during feedback. These simulations suggest that the distinct spatial profile of mnemonic responses in visual cortex may be a straightforward consequence of reversing the flow of information in a system with hierarchical structure and reciprocal connectivity, and that spatial pooling accumulated during feedforward processing may not be inverted during reinstatement. More broadly, these results demonstrate that models of the visual system may be useful for probing the mechanisms that support and constrain visual memory.

## Discussion

In the current work, we combined empirical and modeling approaches to explore how episodic memories are represented in the human visual system. By using computational models of spatial encoding to compare perceptual and mnemonic BOLD activity, we provide evidence that visual memory, like visual perception, produces retinotopically mapped activation throughout visual cortex. Critically, however, we also identified systematic differences in the population spatial tuning properties of perceptual and mnemonic activity. Compared to perceptual responses, mnemonic responses were lower in amplitude in all visual areas. Further, while we observed a three-fold change in spatial precision from early to late visual areas during perception, mnemonic responses violated this pattern. Instead, mnemonic responses displayed consistent spatial precision across visual areas. Notably, simulations showed that neither reduced SNR nor memory failure could account for this difference. We speculate, instead, that this difference arises from a reversal of information flow in a hierarchically organized and reciprocally connected visual cortex. To support this, we show that top-down activation in a simple hierarchical model elicits a systematically different pattern of responses than bottom-up activation. These simulations reproduce the properties we observe during both perception and memory. Together, these results reveal properties of memory-driven activity in visual cortex and suggest specific computational processes governing visual cortical responses during memory retrieval.

Much work in neuroscience has been dedicated to the question of how internally generated stimulus representations are coded in the brain. Early neuroimaging work established that sensory cortices are recruited during imagery and memory tasks[5,6,8], moving the field away from purely symbolic accounts of memory[51]. More recently, memory researchers have favored decoding and pattern similarity approaches over univariate activation analyses to examine the content of retrieved memories[10,11,52]. While these approaches are powerful, they do not explicitly specify the form mnemonic activity should take, and many activation schemes can lead to successful decoding or changes in pattern correlations. In the present work, we leveraged encoding models from visual neuroscience, specifically stimulus-referred pRF models, to examine and account for memory-triggered activity in visual cortex. In contrast to decoding or pattern similarity approaches, encoding models predict the activity evoked in single voxels in response to sensory or cognitive manipulations using a set of explicit mathematical operations[53]. Spatial encoding models have proved particularly powerful because space is coded in the human brain at a scale that is well-matched to the millimeter sampling resolution of fMRI[54–56]. Despite the power of such encoding models, relatively little work has applied these models to questions about long-term memory (though see refs. 15, 16, 57, 58). Here, using this approach, we revealed

properties of memory responses in visual cortex that decoding approaches have missed. Most notably, we found that memory activity is characterized by a different pattern of spatial precision across regions than perceptual activity. Because spatial parameters such as polar angle are explicitly modeled in pRF models, we were able to quantify and interpret these differences.

Because encoding models of space confer such analytical advantages, we specifically designed our fMRI analyses to assess neural coding of spatial location during memory retrieval. The spatial tuning evident in our results and our simulations are consistent with the idea that participants were in fact remembering the stimulus location during the fMRI scan. Although the spatial location was the focus of our analysis, this does not imply that participants only remembered this dimension of the stimuli. Participants were not told that fMRI analyses would focus on the spatial location of the stimulus and were trained to remember both spatial location and pattern features of the stimuli. Hence it is likely that participants were retrieving both location and pattern. Our design does not allow us to demonstrate stimulus pattern reactivation because participants only saw one stimulus pattern per location. However, prior studies have shown that stimulus dimensions including orientation[12,16], spatial frequency[16], shape[59,60], and pattern[49] are encoded in visual cortex activity during memory retrieval. Might pattern tuning differ between perception and memory in a similar manner to spatial location tuning? Answering this question will require developing well-validated encoding models for stimulus pattern that do not currently exist. There has, however, been recent progress in this direction. A recent paper by Breedlove and colleagues[57] demonstrated that tuning for a non-location feature (spatial frequency) differs between perception and memory. Developing visual encoding models that can account for increasingly complex stimulus properties will be a critical component in the effort to understand memory representations.

Our results have implications for the study of memory reactivation. First, our findings suggest that the specific architecture of a sensory system may constrain what memory reactivation looks like in that system. Though memory reactivation is often studied in sensory domains, the architecture of these systems is not usually considered when interpreting reactivation effects. Here, we propose that hierarchical spatial pooling in visual cortex produces a systematic and distinct pattern of memory reactivation that cannot be attributed to memory failure. Accounting for effects like this is necessary for developing a quantitative model of memory reactivation. Further, even when failures of the memory system are of interest, care should be taken to avoid confounding memory failure effects with the kinds of effects described here. Second, our results advocate for shifting the emphasis of memory reactivation research away from similarities with perception to differences between them. Most previous work has focused on identifying similarities between the neural substrates of visual perception and visual memory. These studies have been successful in that they have produced many positive findings of memory reactivation in human visual cortex[5–14]. However, much of this work implicitly assumes that any mismatch between perception and memory is due to the fact that memory reactivation is either inherently low fidelity or susceptible to noise[23], or is a subset of the perceptual response[7,8,61]. Our results demonstrate that, at least in the spatial domain, this is not the case, and that systematic differences beyond noise exist.

Other recent work has also argued for reconsidering a strict view of reactivation. A recently published study using similar methods reported a highly complementary finding to ours: that individual voxels in human visual cortex have larger pRFs during imagery than during perception[57]. While our empirical results and those of Breedlove and colleagues[57] are mutually supportive, our results are also distinct from theirs because we quantify neural responses at the population level rather than the voxel-level and rule out numerous

confounds related to memory task performance. Our results are also broadly consistent with other recent studies reporting computational[62] and neural[63–67] differences between perception and memory responses in the visual system. Ultimately, the field should strive to quantify and explain these differences in order to fully understand the neural basis of memory retrieval.

Despite the usefulness of encoding models like pRF models for quantifying neural responses in a stimulus-referred space, these models do not provide a natural explanation for why perception and memory responses differ. We show in Fig. 6 that pRF models fail to capture the aspects of memory responses that are distinct from perceptual responses: namely, the dramatic change in spatial precision. While it would be possible to fit separate pRF parameters to memory data to improve the ability of the model to accurately predict memory responses, this approach would not explain why these parameters or responses differ. How then can we account for this? We were particularly intrigued by the possibility that differences between memory and perception activity are a consequence of the direction of processing in hierarchically-organized cortex. Hierarchical structure and feedback processing are not typically directly simulated in a pRF model but there is considerable evidence to suggest these factors are of interest. Studies of anatomical connectivity provide evidence that the visual system is organized approximately hierarchically[44,68] (for other perspectives see refs. 69, 70), and that most connections within the visual system are reciprocal[44]. Studies also show that the hippocampus sits atop the highest stage of the visual hierarchy, with reciprocal connections to high-level visual regions via the medial temporal lobe cortex[43–45]. The hippocampus is also strongly coupled to a widespread cortical network, including retrosplenial/posterior cingulate cortex, lateral parietal and temporal cortices, and lateral and medial prefrontal cortices[71,72]. Some of these regions may also exert top-down drive over visual cortex. These observations make the prediction that initial top-down drive from the hippocampus and its cortical network during memory retrieval may result in the backwards propagation of neural activity through the visual system. Neural recordings from the macaque[46] and human[47–49], as well as computational modeling[73] support the idea of backwards propagation.

Based on these observations and our hypothesis, we constructed a hierarchical network model in which we could simulate top-down activity. Though this model shares some features of hierarchical models of object recognition[74,75], our implementation is simpler. Our model is entirely linear, its parameters are fixed (not the result of training), and it encodes only one stimulus feature: space. In contrast to pRF models, which express each voxel's activity as a function of the stimulus, hierarchical models express each layer's activity as a function of the previous layer's activity. While hierarchical models are often run in a feedforward direction, they can be adapted to also incorporate feedback processes[76]. While highly simplified, the simulations we performed captured the dominant features of our data, providing a parsimonious explanation for our observations. These simulations suggest that the greater similarity between perception and memory in later areas than in earlier areas is a consequence of spatial pooling and the reversal of information flow during memory retrieval. Our simulations also indicate that some trends present in our data warrant further investigation. For instance, while we could not conclude that the earliest visual cortical areas had the least precise responses during memory (a reversal of the perception pattern), our simulations predict that this effect should be present, albeit significantly weaker than in the feedforward direction. Our model predictions diverge from those of a recently published generative model of visual imagery[57] in predicting this reversal. Like our model, the Breedlove et al. model predicts overall coarser spatial responses during imagery than during perception. However, whereas the Breedlove et al. model predicts that late visual areas should have coarser spatial responses than early visual areas during both perception and memory, we predict a reversal of the

perception pattern during memory. Given the small sample size of both our study and that of Breedlove et al., and the small size of the expected effect, future work should adjudicate between the predictions of these models with a highly powered experiment.

Our simulations also raise questions and generate predictions about the consequences of visual cortical architecture for cognition. First, why have a hierarchical architecture in which the detailed information present in early layers cannot be reactivated? The hierarchical organization of the visual system is thought to give rise to the low-level feature invariance required for object recognition[74,75]. Our results raise the possibility that the benefits of such an architecture for recognition outweigh the cost of reduced precision in top-down responses. Second, how is it that humans have spatially precise memories if visual cortical responses do not reflect this? One possibility is that read-out mechanisms are not sensitive to all of the properties of mnemonic activity we measured. For instance, memory decisions could be driven exclusively by the neural population with the strongest response (e.g. those at the peak of the polar angle response functions). Another possibility is that reactivation in other non-sensory regions[52,58,77,78], which may express different properties, is preferentially used to guide memory-based behavior. These, and other possibilities should be explored in future work that simultaneously measures brain activity and behavioral memory precision in larger cohorts of participants than were used here.

Sensory reactivation during long-term memory retrieval has clear parallels to sensory engagement in other forms of memory such as iconic memory and working memory. Nonetheless, there may also be differences in the specific way that sensory circuits are activated across these forms of memory. One critical factor may be how recently the sensory circuit was activated by a stimulus at the time of memory retrieval. In iconic memory studies, very detailed information can be retrieved if probed within a second of the sensory input[79]. In working memory studies, sensory activity is thought to be maintained by active mechanisms through a seconds-long delay[33]. In imagery studies, eye-specific circuits presumed to be in V1 can be re-engaged if there is a delay of 5 min or less from when the participant viewed stimuli through the same eye, but not if there is a delay of 10 min[25]. Long-term forms of episodic memory retrieval and imagery are both thought to be capable of driving visual activity at much longer delays. While episodic memory retrieval depends on the hippocampus, some forms of imagery may not require hippocampal processing at all[80]. Given that the mechanism for engaging sensory cortex may differ across these different forms of memory, the question of how similar sensory activation is across these timescales remains an open question.

The field of visual working memory in particular has relied on very similar methods to the ones we use here to investigate the role of visual cortex in memory maintenance[39,81]. Many such studies have shown that early visual areas contain retinotopically specific signals throughout a delay period[34,39,82]. These studies agree with ours in that they demonstrate a role for visual cortex in representing mnemonic information, and establish that some properties of neural coding during perception are preserved during memory. However, to the best of our knowledge, no working memory study has reported precision differences between perceptual and mnemonic activity akin to what we report. Informal assessment of recent working memory papers suggests that stimulus reconstructions made from working memory delay period activity are approximately as precise as reconstructions made from perceptual activity[34,35,39]. There are theoretical reasons to expect a difference between working memory and episodic memory representations. During typical working memory tasks, a visual cortical representation that was just evoked must be kept activated. However, during episodic memory tasks such as the paired associates task we use here, there is no recently evoked representation of the stimulus in visual cortex. This representation must be created anew, and in standard models of memory retrieval, this process is initiated by the hippocampus and its

cortical network. As described by our model, this mechanism for reactivating visual cortex may come with unique costs relative to working memory maintenance. Ultimately, a direct comparison between working memory and episodic memory in the same experiment will be required to make progress on this question.

Our results raise questions about whether long-term memory and endogenous attention share mechanisms for modulating the response of visual cortical populations. Both attention and memory can activate visual cortex. Spatial and feature-based attention have been shown to enhance neural responses to stimuli[83–88]. Similar to our results, attention has also been shown to enhance visual cortical activation in the absence of any visual stimulation[89–94]. Given these findings, are the responses we observed during memory retrieval better characterized as long-term memory reactivation or as attention? There are a number of reasons why we don't think it's useful to attribute our findings to attention alone. First, characterizing our results as spatial attention does not explain them. We report that the precision of spatial tuning during memory retrieval differs from the precision of spatial tuning during perception. To the best of our knowledge, there is no parallel finding in the spatial attention literature. Second, there are meaningful differences between our task and typical endogenous attention tasks. Most attention tasks have no memory component since the cue explicitly represents the attended location or feature. In contrast, in our task, the correct spatial location or pattern cannot be determined from the fixation cue without having previously encoded the association between them and then successfully recalling it. Thus, our task by necessity involves memory. It may also involve attention, to the extent that items retrieved from memory become the targets of internal attention[95]. From our perspective, it is plausible that the neural responses we observed during memory retrieval resemble those observed when visual attention is deployed in the absence of a stimulus. Future experiments should address this question directly. Further, modeling efforts should address whether it's possible to develop a model of top-down processing in visual cortex that can account for both memory retrieval and attention, or whether separate computational models are needed.

In the current work, we provide empirical evidence that memory retrieval elicits systematically different activation in human visual cortex compared to visual perception. Using simulations and a network model of cortex, we argue that these distinctions arise from a reversal of information flow within a hierarchically structured visual system. Collectively, this work makes progress on providing a detailed account of reactivation in visual cortex and sheds light on the broader computational principles that guide top-down processes in sensory systems.

## Methods
### Participants
Nine human participants participated in the experiment (5 males, 22–46 years old), following procedures approved by the New York University Institutional Review Board. All participants had normal or correct-to-normal visual acuity, normal color vision, and no MRI contraindications. Participants were recruited from the New York University community and included author S.E.F. and author J.W. All participants gave written informed consent prior to participation. Participants were compensated $30/h for participation. Participants who were NYU employees and volunteered to perform the study during work hours waived compensation, as approved by the New York University Institutional Review Board. No participants were excluded from data analysis.

### Stimuli
Experimental stimuli included nine unique radial frequency patterns (Fig. 1a). We first generated patterns that differed along two dimensions: radial frequency and amplitude. We chose stimuli that tiled a one-dimensional subspace of this two-dimensional space, with radial

frequency inversely proportional to amplitude. The nine chosen stimuli took radial frequency and amplitude values of: [2, 0.9], [3, 0.8], [4, 0.7], [5, 0.6], [6, 0.5], [7, 0.4], [8, 0.3], [9, 0.2], [10, 0.1]. We selected four of these stimuli to train participants on in the behavioral training session and to appear in the fMRI session. For every participant, those stimuli were: [3, 0.8], [5, 0.6], [7, 0.4], [9, 0.2]; (radial frequency, amplitude). The remaining five stimuli were used as lures in the test trials of the behavioral training session. Stimuli were generated using a publicly available script[96], saved as images, and cropped to the same size.

### Experimental procedure
The experiment began with a behavioral training session, during which participants learned four paired associates (Fig. 1). Specifically, participants learned that four colored fixation dot cues were uniquely associated with four spatially localized radial frequency patterns. An fMRI session immediately followed completion of the behavioral session (Fig. 2a). During the scan, participants completed two types of functional runs: (1) perception, where they viewed the cues and associated spatial stimuli; and (2) memory, where they viewed only the fixation cues and recalled the associated spatial stimuli. Details for each of these phases are described below. A separate retinotopic mapping session was also performed for each participant (Fig. 2b), which is described in the next section.

**Behavioral training.** For each participant, the four radial frequency patterns were first randomly assigned to one of four polar angle locations in the visual field (45°, 135°, 225°, or 315°) and to one of four colored cues (orange, magenta, blue, green; Fig. 1b). Immediately before the fMRI session, participants learned the association between the four colored cues and the four spatially localized stimuli through interleaved study and test blocks (Fig. 1c). Participants alternated between study and test blocks, completing a minimum of four blocks of each type. Participants were required to reach at least 95% accuracy, and performed additional rounds of study-test if they did not reach this threshold after four test blocks. The training task was implemented in PsychoPy v1.85.6[97].

During study blocks, participants were presented with the associations. Participants were instructed to maintain central fixation and to learn each of the four associations in anticipation of a memory test. At the start of each study trial (Fig. 1c), a central white fixation dot (radius = 0.1 dva) switched to one of the four cue colors. After a 1 s delay, the associated radial frequency pattern appeared at 2° of eccentricity and its assigned polar angle location in the visual field. Each pattern image subtended 1.5 dva and was presented for 2 s. The fixation dot then returned to white, and the next trial began after a 2 s interval. No participant responses were required. Each study block contained 16 trials (4 trials per association), presented in random order.

During test blocks, participants were presented with the colored fixation dot cues and tested on their memory for the associated stimulus pattern and spatial location. Participants were instructed to maintain central fixation and to try to covertly recall each stimulus when cued, and then to respond to the test probe when prompted. At the start of each test trial (Fig. 1c), the central white fixation dot switched to one of the four cue colors. This cue remained on the screen for 2.5 s while participants attempted to covertly retrieve the associated stimulus. At the end of this period, a test stimulus was presented at 2° of eccentricity for 2 s. Then, participants were cued to make two consecutive responses to the test stimulus: whether it was the correct radial frequency pattern (yes/no) and whether it was presented at the correct polar angle location (yes/no). Test stimuli were presented at a random orientation on each trial. Each test stimulus had a 50% probability of being the correct pattern. Incorrect patterns were drawn randomly from the three patterns associated with other cues and the

five lure patterns (Fig. 1a). Each test stimulus had a 50% probability of being in the correct polar angle location, which was independent from the probability of being the correct pattern. Incorrect polar angle locations were drawn from the three locations assigned to the other patterns and 20 other evenly spaced locations around the visual field (Fig. 1a). This placed the closest spatial lure at 15° of polar angle away from the correct location. Responses were solicited from the participant with the words "Correct pattern?" or "Correct location?" displayed centrally in white text. The order of these queries was counterbalanced across test blocks. Participants responses were recorded on a keyboard with a maximum response window of 2 s. Immediately after a response was made or the response window closed, the color of the text turned black to indicate an incorrect response if one was made. After this occurred for both queries, participants were presented with the colored fixation dot cue and correct spatially localized pattern for 1 s as feedback. This feedback occurred for every trial, regardless of participant responses to the probe. Each test block contained 16 trials (4 trials per association), presented in random order.

**fMRI session.** During the fMRI session, participants performed two types of functional runs: perception and memory retrieval (Fig. 2a). Participants completed 5–6 runs (~3.5 min each) of both perception and memory tasks in an interleaved order. This amounted to 40–48 repetitions of perceiving each stimulus and of remembering each stimulus per participant. Both perception and memory tasks were implemented in PsychoPy v1.85.6[97].

During perception runs, participants viewed the colored fixation dot cues and the radial frequency patterns in their learned locations. Participants were instructed to maintain central fixation and to perform a one-back task on the stimuli. The purpose of the one-back task was to encourage covert stimulus-directed attention on each trial. At the start of each perception trial (Fig. 2a, top), a central white fixation dot (radius = 0.1 dva) switched to one of the four cue colors. After a 0.5 s delay, the associated radial frequency pattern appeared at 2° of eccentricity and its assigned polar angle location in the visual field. Each pattern subtended 1.5 dva and was presented for 2.5 s. The fixation dot then returned to white and the next trial began after a variable interval. Intervals were drawn from an approximately geometric distribution sampled at 3, 4, 5, and 6 sec with probabilities of 0.5625, 0.25, 0.125, and 0.0625, respectively. Participants indicated when a stimulus repeated from the previous trial using a button box. Responses were accepted during the stimulus presentation or during the interstimulus interval. Each perception run contained 32 trials (8 trials per stimulus). The trial order was randomized for each run, separately for every participant.

During memory runs, participants viewed the colored fixation dot cues and recalled the associated patterns in their learned spatial locations. Participants were instructed to maintain central fixation, to use the cues to initiate recollection, and to make a subjective judgment about the vividness of their memory on each trial. The purpose of the vividness task was to enforce attention to the remembered stimulus on each trial. At the start of each memory trial (Fig. 2a, top), the central white fixation dot switched to one of the four cue colors. This cue remained on the screen for a recollection period of 3 sec. The fixation dot then returned to white and the next trial began after a variable interval. Participants indicated whether the stimulus associated with the cue was vividly remembered, weakly remembered, or not remembered using a button box. Responses were accepted during the cue presentation or during the interstimulus interval. Each memory run contained 32 trials (8 trials per stimulus). For a given participant, each memory run's trial order and trial onsets were exactly matched to one of the perception runs. The order of these matched memory runs was scrambled relative to the order of the perception runs.

## Retinotopic mapping procedure

Each participant completed either 6 or 12 identical retinotopic mapping runs in a separate fMRI session from the main experiment (Fig. 2b, top). Stimuli and procedures for the retinotopic mapping session were based on those used by the Human Connectome Project[36] and were identical to those reported by Benson and Winawer[98]. During each functional run, bar apertures on a uniform gray background swept across the central 24 degrees of the participant's visual field (circular aperture with a radius of 12 dva). Bar apertures were a constant width (1.5 dva) at all eccentricities. Each sweep began at one of eight equally spaced positions around the edge of the circular aperture, oriented perpendicularly to the direction of the sweep. Horizontal and vertical sweeps traversed the entire diameter of the circular aperture while diagonal sweeps stopped halfway and were followed by a blank period. A full-field sweep or half-field sweep plus blank period took 24 s to complete. One functional run contained 8 sweeps, taking 192 s in total. Bar apertures contained a grayscale pink noise background with randomly placed faces, scenes, objects, and words at a variety of sizes. Noise background and stimuli were updated at a frequency of 3 Hz. Each run of the task had an identical design. Participants were instructed to maintain fixation on a central dot and to use a button box to report whenever the dot changed color. Color changes occurred on average every 3 s.

## MRI acquisition

Images were acquired on a 3T Siemens Prisma MRI system at the Center for Brain Imaging at New York University. Functional images were acquired with a T2*-weighted multiband EPI sequence with whole-brain coverage (repetition time = 1 s, echo time = 37 ms, flip angle = 68°, 66 slices, 2 × 2 × 2 mm voxels, multiband acceleration factor = 6, phase-encoding = posterior-anterior) and a Siemens 64-channel head/neck coil. This sequence was based on the CMRR MultiBand Accelerated EPI Pulse Sequences (Release R015a)[99–101]. Spin echo images with anterior-posterior and posterior-anterior phase-encoding were collected to estimate the susceptibility-induced distortion present in the functional EPIs. Between one and three whole-brain T1-weighted MPRAGE 3D anatomical volumes (0.8 × 0.8 × 0.8 mm voxels) were also acquired for seven participants. For two participants, previously acquired MPRAGE volumes (1 × 1 × 1 mm voxels) from a 3T Siemens Allegra head-only MRI system were used.

## MRI processing

**Preprocessing.** Anatomical and functional images were preprocessed using FSL v5.0.10[102] and Freesurfer v5.3.0[103] tools implemented in nipype v1.1.9 workflow[104]. To correct for head motion, each functional image acquired in a session was realigned to a single band reference image and then registered to the spin echo distortion scan acquired with the same phase encoding direction. The two spin echo images with reversed phase encoding were used to estimate the susceptibility-induced distortion present in the EPIs. For each EPI volume, this non-linear unwarping function was concatenated with the previous spatial registrations and applied with a single interpolation. Freesurfer was used to perform segmentation and cortical surface reconstruction on each participant's average anatomical volume. Registration from the functional images to each participant's anatomical volume was performed using boundary-based registration. Preprocessed functional time series were then projected onto each participant's reconstructed cortical surface.

**GLM analyses.** Beginning with each participant's surface-based time series, we used GLMdenoise v1.4[105] to estimate the neural pattern of activity evoked by perceiving and remembering every stimulus (Fig. 2a). GLMdenoise improves signal-to-noise ratios in GLM analyses by identifying a pool of noise voxels whose responses are unrelated to

the task and regressing them out of the time series. This technique first converts all time series to percent signal change and determines an optimal hemodynamic response function for all vertices using an iterative linear fitting procedure. It then identifies noise vertices as vertices with negative $R^2$ values in the task-based model. Then, it derives noise regressors from the noise pool time series using principal components analysis and iteratively projects them out of the time series of all vertices, one noise regressor at a time. The optimal number of noise regressors is determined based on cross-validated $R^2$ improvement for the task-based model. We estimated two models using this procedure. We constructed design matrices for the perception model to have four regressors of interest (one per stimulus), with events corresponding to stimulus presentation. Design matrices for the memory model were constructed the same way, with events corresponding to the cued retrieval period. These models returned parameter estimates reflecting the BOLD amplitude evoked by perceiving or remembering a given stimulus versus baseline for every vertex on a participant's cortical surface (Fig. 2a, bottom).

**Fitting pRF models.** Images from the retinotopic mapping session were preprocessed as above, but omitting the final step of projecting the time series to the cortical surface. Using these time series, non-linear symmetric 2D Gaussian population receptive field (pRF) models[29,31] were estimated in Vistasoft v1.0 (Fig. 2b). We refer to this nonlinear version of the pRF model as the compressive spatial summation (CSS) model, following Kay and colleagues[31]. Briefly, we estimated the receptive field parameters that, when applied to the drifting bar stimulus images, minimized the difference between the observed and predicted BOLD time series. First, stimulus images were converted to contrast apertures and downsampled to $101 \times 101$ grids. Time series from each retinotopy run were resampled to anatomical space and restricted to gray matter voxels. Time series were then averaged across runs. pRF models were solved using a two-stage coarse-to-fine fit on the average time series. The first stage of the model fit was a coarse grid fit, which was used to find an approximate solution robust to local minima. This stage was solved on a volume-based time series that was first temporally decimated, spatially blurred on the cortical surface, and spatially subsampled. The parameters obtained from this fit were interpolated and then used as a seed for subsequent nonlinear optimization, or fine fit. This procedure yielded four final parameters of interest for every voxel: eccentricity ($r$), polar angle ($\theta$), sigma ($\sigma$), exponent ($n$). The eccentricity and polar angle parameters describe the location of the receptive field in space, the sigma parameter describes the size of the receptive field, and the exponent describes the amount of compressive spatial summation applied to responses from the receptive field. Eccentricity and polar angle parameters were converted from polar coordinates ($r$, $\theta$) to rectangular coordinates ($x$, $y$) for some analyses. Variance explained by the pRF model with these parameters was also calculated for each voxel. All parameters were then projected from each participant's anatomical volume to the cortical surface (Fig. 2b, bottom).

**ROI definitions**
Regions of interest were defined by hand-drawing boundaries between visual field maps on each participant's cortical surface. For each map, we drew boundaries at polar angle reversals following established practice[106]. We used this method to define six ROIs spanning early to mid-level visual cortex: V1, V2, V3, hV4, LO (LO1 and LO2), and V3ab (V3a and V3b). We had several goals in mind when choosing these ROIs. First, we wanted ROIs that spanned the visual hierarchy and that were known to have different receptive field sizes. Second, we wanted to avoid the most anterior maps, which require large quantities of data to define reliably in individual participants. We selected early visual areas V1-V3, based on their well-described anatomical boundaries and organization in

humans[106]. We then selected the visual areas bordering V3 from each of the ventral, lateral, and dorsal streams.

We further restricted each ROI by preferred eccentricity in order to isolate vertices responsive to our stimuli. We excluded vertices with eccentricity values $<0.5°$ and $>8°$. This procedure excluded vertices responding primarily to the fixation dot and vertices near the maximal extent of visual stimulation in the scanner. We also excluded vertices whose variance explained by the pRF model ($R^2$) was $<0.1$, indicating poor spatial selectivity. All measures used to exclude vertices from ROIs were independent of the measurements made during the perception and memory tasks.

**Analyses quantifying perception and memory activity**
**2D visualizations.** Our main analyses examined the BOLD response evoked by perceiving and remembering the experimental stimuli as a function of visual field parameters estimated from the pRF model. Our first step was to visualize evoked activity during perception and memory in visual field coordinates (Fig. 3a). Transforming the data in this way allowed us to view the activity in a common reference frame across all brain regions, rather than on the cortical surface, where comparisons are made difficult by the fact that surface area and cortical magnification differ substantially from one area to the next. To do this, we selected the ($x$, $y$) parameters for each surface vertex from the retinotopy model and the $\beta$ parameters from the GLM analysis. Separately for a given ROI, participant, stimulus, and task (perception/memory), we interpolated the $\beta$ values over ($x$, $y$) space. We rotated each of these representations according to the polar angle location of the stimulus so that they would be aligned at the upper vertical meridian. We then z-scored each representation before averaging across stimuli and participants. We used these images to gain intuition about the response profiles and to guide subsequent quantitative analyses.

**Polar angle response functions.** Before quantifying these representations, we simplified them further. Because our stimuli were all presented at the same eccentricity, we reduced our 2D stimulus coordinate representations to 1D dimensional responses functions on the polar angle dimension (Fig. 4a). We did this by selecting surface vertices whose pRFs were within one $\sigma$ of the stimulus eccentricity (2°) for each ROI. We then binned the vertices into 18 bins of polar angle distance from the stimulus and took the median evoked BOLD response within each bin to produce polar angle response functions for each participant. We divided each participant's response function by the norm of the response vector before taking the mean across participants and then multiplying by the average vector norm to get the correct units back. This procedure prevents a participant with a high BOLD response across all polar angles from dominating the average response. The resulting average polar angle response functions showed clear surround suppression for polar angles near the stimulus during perception. Given this, we fit a difference of two von Mises distributions to the average data, with the location parameters ($\mu$) for the two von Mises distributions fixed to be equal, but the spread ($\kappa_1$, $\kappa_2$) and scale ($\beta_1$, $\beta_2$) allowed to differ. Note that these are different $\beta$ parameters than those estimated from GLMdenoise, which we refer to above.

**Group-level quantification.** We quantitatively assessed the similarities and differences between perception and memory responses using these difference of von Mises fits. We interpreted the value of the location parameter as a neural indicator of the perceived or remembered polar angle location. We also computed an amplitude metric that quantified the height of the fit difference of von Mises (max - min). This measure indicates the relative strength of the BOLD response in the vertices most responsive to the stimulus. Finally, we computed the FWHM of the fit difference of Von Mises as an indicator of the precision of the BOLD response, or spread of the response to vertices that code

for polar angles away from the stimulus. We repeated our across-participant averaging and von Mises fitting procedure 500 times, drawing participants with replacement, to create bootstrapped 68% and 95% confidence intervals for both perception and memory location, amplitude, and FWHM parameters. We used these confidence intervals to make inferences about spatial tuning differences between perception and memory and across visual areas.

**Individual participant quantification.** We estimated the same location, amplitude, and FWHM parameters using data from individual participants. We first computed average polar angle response functions for each participant as described in the prior section. Because participants varied in how offset these response functions were from 0% signal change, which our group-level von Mises fitting procedure was not designed to account for, we removed these offsets by shifting the response functions such that vertices farthest away from the stimulus (−160°, 160°, and 180°) had a mean response of 0%. We then fit a difference of von Mises to each participant's response function. In addition to extracting the location, amplitude, and FWHM measures described in the prior section for each participant, we also computed the $R^2$ of the fit difference of von Mises (Supplementary Fig. 2).

We re-assessed the main effects of ROI, the main effects of perception vs memory, and the interaction of these variables on location, amplitude, and FWHM values using individual participant values. To do this, we entered individual participant parameters into repeated measures ANOVAs. In all models, ROI was coded numerically according to hypothesized position in the visual hierarchy (V1 = 1; V2 = 2; V3 = 3; hV4 = 4; LO = 5; V3ab = 6) and perception/memory as a categorical variable. We report $F$ values from these models and two-tailed $p$ values under $\alpha = 0.05$. Because of uncertainty about how cleanly visual field areas map onto such a hierarchy, we recomputed these statistics with ROI coded as a categorical variable. All inferences remained the same. We assessed whether data met the assumptions for an ANOVA by evaluating normality and sphericity. We determined that the data met assumptions for normality by visualizing the model residuals using q-q plots and by using the Shapiro-Wilk test. We determined that the data met assumptions for sphericity using Mauchly's test, though this value was close to significance in some cases. Inferences were the same when the Huynh-Feldt correction for violations of sphericity was applied.

**Eccentricity response functions.** We performed a cursory evaluation of eccentricity-dependent BOLD responses during perception and memory. For each stimulus, we first excluded all vertices further than 15° of polar angle from the stimulus. We then divided vertices into eccentricity bins of 0.5 dva (min = 0.5 dva and max = 8 dva). For each participant, task, and ROI, we computed the median evoked BOLD response within each bin, and averaged this value across participants as described in the section above on polar angle response functions. Because these eccentricity response functions were asymmetric and noisy, we did not fit a parametric function to these binned data. Instead, we simply plot the binned estimates, with linear interpolation between adjacent bins (Supplementary Fig. 1). We obtained 95% confidence intervals for each bin by resampling participants with replacement 500 times, and recomputing the bin averages.

**Software.** Statistical quantification and data visualizations for these analyses and all those subsequently described were made using nibabel 3.2.1[107], numpy 1.21.2[108], scipy 1.7.1[109], pandas 1.3.3[110,111], matplotlib 3.4.3[112], and seaborn 0.11.2[113].

**Noise simulations**
We performed four simulations designed to test whether differences in noise between perception and memory data could explain differences in the responses we observed. We identified four potential types of

noise that were present in our memory data but not our perception data: (1) reduced SNR; (2) retrieval task lapses; (3) associative memory errors; (4) angular memory errors. We then simulated the effect of these types of noise on our perception data and asked whether these noise sources could produce responses similar to the ones we observed during memory.

**SNR simulation.** To simulate reduced SNR, we created artificial datasets with different amounts of additive noise introduced to every vertex's perception parameter estimate. Noise was added in five levels: noise needed to regenerate the empirical SNR of the perception data (p), noise needed to generate the empirical SNR of the memory data (m), or noise needed to generate 1/2, 1/4, or 1/8 the empirical SNR of the memory data. For each of these values, we simulated 100 independent datasets for every participant and ROI. We determined the amount of signal and noise actually observed for each vertex during perception and memory by examining bootstrapped parameter estimate distributions produced by GLMdenoise. We defined the median parameter estimate across bootstraps as the amount of signal and the standard error of this distribution as the amount of noise. To simulate new data for a vertex, we randomly drew a new parameter estimate from a normal distribution defined by the true signal value (median) and the noise value (SE) needed to produce the target SNR. Critically, we made the draws correlated across vertices for each simulation. We did this by selecting a scale factor from a standard normal distribution which determined how many SEs away from the median every vertex's simulated value would lie. This scale factor was shared across all vertices in an ROI for a given simulation. This procedure overcompensates for the spatial correlation present in BOLD data by assuming that SNR is 100% correlated across all vertices in an ROI. Note that if the noise were uncorrelated across vertices, it would have a much smaller effect on the population tuning curves. For each noise value and each of the 100 simulations, we analyzed the simulated data using the same procedure we applied to the actual data. This yielded 100 von Mises fits to the simulated data for each noise value and ROI (Supplementary Fig. 3a). We extracted the location, amplitude, and FWHM values from these fits. For very high noise simulations, von Mises fitting sometimes failed. We evaluated whether the location and FWHM values approximated the ones we observed during memory by calculating the proportion of simulations that fell within the 95% confidence intervals derived from the memory data (Fig. 5a and Supplementary Fig. 4a).

**Retrieval task lapse simulation.** To simulate retrieval task lapses, we created artificial datasets that contained a variable number of perception trials with no signal. Retrieval task lapse was simulated in five levels: 0%, 25%, 50%, 75%, and 100% of trials. For each of these values, we simulated 100 independent datasets for every participant and ROI. Depending on the percentage of lapses, zero, one, two, three, or all four stimuli were randomly designated as "lapsed" in each simulated dataset. For the lapsed stimuli, new parameter estimates were drawn from a distribution defined by zero signal during perception for every vertex. For the remaining stimuli, new parameter estimates were drawn from a distribution defined by the true perception signal for every vertex. The noise was equated for both trial types; for each vertex, we used the amount of noise observed during perception. We performed this analysis at the level of stimuli, rather than trials because our GLM yielded parameter estimates for each stimulus, not each trial. As in the SNR simulation, simulated data were correlated across vertices in an ROI and simulated data were analyzed using the same procedures as for the actual data, yielding von Mises fits (Supplementary Fig. 3b). For the highest lapse rates, von Mises fitting sometimes failed. We evaluated whether simulated location and FWHM values approximated the ones we observed during memory by

calculating the proportion of simulations that fell within the 95% confidence intervals derived from the memory data (Fig. 5b and Supplementary Fig. 4b).

**Associative memory error simulation.** To simulate associative memory errors, we created artificial datasets that contained a variable number of perception trials where the response corresponded to one of the other studied stimuli. Associative memory errors were simulated in four levels: 0%, 25%, 50%, 75%, and 100% of trials. For each of these values, we simulated 100 independent datasets for every participant and ROI. Depending on the percentage of errors, zero, one, two, three, or all four stimuli were randomly designated as "associative errors" in each simulated dataset. For error stimuli, one of the three other studied stimuli was randomly chosen and new parameter estimates were drawn from this distribution rather than the correct one. For the remaining stimuli, new parameter estimates were drawn from a distribution defined by the true perception signal for every vertex. As in the task lapse simulation, we performed this analysis at the level of stimuli, rather than trials because our GLM yielded parameter estimates for each stimulus, not each trial. Like the prior simulations, simulated data were correlated across vertices in an ROI and simulated data were analyzed using the same procedures as for the actual data, yielding von Mises fits (Supplementary Fig. 3c). For the highest error rates, von Mises fitting sometimes failed. We evaluated whether simulated location and FWHM values approximated the ones we observed during memory by calculating the proportion of simulations that fell within the 95% confidence intervals derived from the memory data (Fig. 5c and Supplementary Fig. 4c).

**Angular memory error simulation.** To simulate angular memory error, we created artificial datasets that contained a variable amount of angular error in the peak location of the perception polar angle response functions. Angular memory error was simulated in five levels of standard deviation: 0, 30, 60, 90, and 180 degrees. For each of these values, we simulated 100 independent datasets for every participant and ROI. We assigned the amount of memory error for a given participant and stimulus by drawing a random value from a normal distribution centered at the true angular location of the stimulus and with the current standard deviation. We then used these memory error values to misalign simulated perception data. Specifically, we created new perception datasets based on the true signal and noise characteristics of our perception data (equivalent to SNR simulation with p noise, 0% retrieval task lapse simulation, or 0% associative error simulation). As in all other simulations, simulated data were correlated across vertices in an ROI, and simulated data were analyzed according to the same procedure as for the actual data. Before averaging the simulated data across stimuli and participants, we rotated each response by the chosen memory error value rather than by the location of that stimulus. That is, instead of rotating the response to a 45° stimulus by 45° to align all stimuli at 0° (as we did in our main analysis), we rotated the response by a value either close to 45 (generated using small standard deviations, representing small errors) or potentially quite far away from 45 (generated using large standard deviations, representing large errors). After averaging and yielding von Mises fits (Supplementary Fig. 3d), we extracted location and FWHM values. For very high standard deviations, von Mises fitting sometimes failed. We then evaluated whether simulated location and FWHM values approximated the ones we observed during memory by calculating the proportion of simulations that fell within the 95% confidence intervals derived from the memory data (Fig. 5d and Supplementary Fig. 4d).

**pRF forward model**
We evaluated the ability of our pRF model to account for our perception and memory measurements. To do this, we used our pRF model as a forward model. This means that we took the pRF model

parameters fit to fMRI data from the retinotopy session (which used a drifting bar stimulus) and used them to generate predicted BOLD responses to our four experimental stimuli. The model takes processed stimulus images as input, and for each of these images, outputs a predicted BOLD response (in units of % signal change) for every cortical surface vertex. Before running the model, we transformed our experimental stimuli into binary contrast apertures with values of 1 where the stimulus was and values of 0 everywhere else. These images were downsampled to the same resolution as the images used to fit the pRF model (101 × 101).

**Model specification.** The pRF forward model has two fundamental operations. In the first operation, a stimulus contrast aperture image is multiplied by a voxel's pRF. In the CSS and linear models, this pRF is defined as a circular symmetric 2D Gaussian, parameterized by a location in the visual field $(x, y)$ and a size $(\sigma)$. In the DoG + CSS version of the model, this pRF is defined as the difference of two such Gaussians, centered at the same location (see next paragraph). The second operation applies a power-law exponent $(n)$ to the result of the multiplication, effectively boosting small responses. This nonlinear operation is the key component of the CSS model and improves model accuracy in high-level visual areas that are known to exhibit subadditive spatial summation[31,114]. The values of the exponent range from 0 to 1, where a value of 1 returns the model to linear. The output of this nonlinear stage is multiplied by a final scale parameter $(\beta)$, which returns the units to % signal change (Fig. 6a).

Because we observed negative surround responses in V1–V3 during perception, we focused mainly on the results of the DoG + CSS model. Prior work has shown that difference-of-Gaussians (DoG) pRF models can account for the center-surround structure we observed[38]. In order to construct DoG pRFs, we converted each pRF from the CSS model we fit to the retinotopy data to a DoG pRF. We chose this approach after encountering difficulty in fitting a DoG pRF model to the retinotopy data. First, we took every 2D Gaussian pRF from the CSS model, and we subtracted from it a second 2D Gaussian pRF that was centered at the same location but was twice as wide and half as high. This ratio of 2σ and 0.5β between the negative and positive Gaussians was fixed for all voxels. In order to prevent the resulting DoG pRF from being systematically narrower and lower in amplitude than the original pRF, we rescaled the σ and β of the original pRF before converting it to a DoG. We multiplied the original σ by $\sqrt{2}$ and the original β by 2, resulting in a DoG pRF with equivalent FWHM and amplitude as the original pRF. Thus, the DoG pRF differed from the original pRF only in the presence of a suppressive surround.

**Evaluating model predictions.** We evaluated the predictions of the DoG+CSS model for our experimental stimuli. After generating a prediction for each participant, stimulus, and surface vertex, we carried these predictions forward through the same analysis pipeline used to analyze our task-based data. This generated predicted polar angle response functions for each ROI (Fig. 6b). We conducted the same procedure on the bootstrapped datasets. Finally, we generated predictions for two simpler pRF models: the CSS model without the DoG pRF shape and a linear model with no exponent parameter (Supplementary Fig. 5a).

We evaluated how well the DOG+CSS model predictions matched our perception versus memory measurements. We extracted location, amplitude, and FWHM measures from the predicted polar angle response functions. We then compared the predicted amplitude and FWHM parameters for each ROI with the actual perception and memory parameters. We evaluated these relationships by fitting a linear model to the predicted versus observed observations. To generate confidence intervals on these fits, we fit a linear model between the predicted parameters and the actual perception/memory

parameters for each of the bootstrapped datasets (Fig. 6c). We repeated the same procedure using predictions from the alternative pRF models as well (Supplementary Fig. 5b).

As another measure of goodness-of-fit, we calculated the coefficient of determination ($R^2$) for the predicted polar angle response functions from the DoG + CSS model and the observed perception and memory polar angle response functions (Supplementary Fig. 6, top). Under this measure, a model that predicts the mean observed response for every value of polar angle distance will have an $R^2$ of zero, with better models producing positive values and worse models producing negative values. We generated confidence intervals for these accuracies by computing $R^2$ values for each of the 500 bootstrapped perception and memory datasets and the yoked pRF predictions. Given the large negative $R^2$ values we observed for our memory data, we considered the extent to which rescaling the predictions would yield better results. We repeated our $R^2$ calculations after rescaling the predicted polar angle response functions to best fit the memory data (Supplementary Fig. 6, bottom). We rescaled the predictions by the single best fitting scale factor across all ROIs. For the sake of comparison between perception and memory, we also separately rescaled the predictions by the single best fitting scale factor for the perception data.

### Hierarchical network model

We assessed whether a simple instantiation of a single neural network model could account for both the perception and memory data. We implemented a fully linear hierarchical model of neocortex in which the activity from each layer was created by pooling activity from the previous layer. This model encodes 1D space only and its parameters are fixed (i.e., it is not trained). For the feedforward simulation, we began with a 1D boxcar stimulus, which was centered at $0°$ and spanned $15°$ of polar angle. We created a fixed Gaussian convolution kernel ($\mu = 0°$, $\sigma = 15°$), which we convolved with the stimulus to create the activity in layer 1. This layer 1 activity was convolved with the same Gaussian kernel to create the layer 2 activity, and this process was repeated recursively for 8 layers (Fig. 7a, left). In order to simulate memory-evoked responses in this network, we made two assumptions. First, we assumed that the feedback simulation began with the layer 8 activity from the feedforward simulation. That is, we assumed no information loss or distortion between perception and memory in the last layer. Second, we assumed that all connections were reciprocal and thus that the same Gaussian kernel was applied to transform layers in the feedback direction as in the feedforward direction (Fig. 7a, right). Thus, in the feedback simulation, we convolved the layer 8 activity with the Gaussian kernel to produce the layer 7 activity and repeated this procedure recursively, ending at layer 1. Note that these computations can be performed with matrix multiplication rather than convolution by converting the convolutional kernel to a Toeplitz matrix, which is how we implemented it. In this case, the transpose of the Toeplitz matrix (itself, as it is symmetric) is used in the feedback direction. We plotted each layer's activation (Fig. 7b) and extracted the location, amplitude and FWHM for each layer using the same procedure we performed on our data (Fig. 7c).

In order to evaluate the effect of our parameter choices on our results, we performed a suite of simulations using different combinations of stimulus size, kernel size, and number of layers. To explore the effect of number of layers, we simulated our base model (described above; stimulus = $15°$; kernel $\sigma = 15°$, number of layers = 8) with 4, 6, and 10 layers and then plotted the FWHM for each layer in the feedforward and feedback directions (Supplementary Fig. 7a). In order to explore the interaction between stimulus size and kernel size, we simulated 16 8-layer models. These models evaluated every combination of stimulus size [$15°$, $30°$, $45°$, $60°$] and kernel size [$\sigma = 5°$, $15°$, $30°$, $45°$]. For each model, we plot the FWHM for each layer in the feedback and feedback directions (Supplementary Fig. 7b).

### Reporting summary

Further information on research design is available in the Nature Research Reporting Summary linked to this article.

## Data availability

Preprocessed MRI data, BOLD activation maps, regions of interest, and behavioral data are deposited on the Open Science Framework at https://osf.io/wc7zy/[115]. Raw MRI data have not been made available due to size constraints, but are available on request from the first author. Source data are provided with this paper.

## Code availability

Analysis code is available on the same OSF page as the data https://osf.io/wc7zy/[115] and can also be found at https://github.com/sfavila/Favila_NatComm_2022.

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

## Acknowledgements
This work was supported by R01 EY027401, R01 EY027964, and R01 MH111417 awarded to J.W. S.E.F. was supported by NIH Blueprint D-SPAN Award F99 NS105223.

## Author contributions
Conceptualization, S.E.F., B.A.K., and J.W.; Methodology, S.E.F. and J.W.; Software, S.E.F. and J.W.; Investigation, S.E.F.; Writing-Original Draft, S.E.F. and J.W.; Writing-Review & Editing, S.E.F., B.A.K., and J.W.; Funding Acquisition, S.E.F. and J.W.; Supervision, B.A.K. and J.W.

## Competing interests
The authors declare no competing interests.
