## [Peer Review File · Nature Communications]

Perception and memory have distinct spatial tuning properties in human visual cortexReviewers' Comments:

Reviewer #1:

Remarks to the Author:

The study reported in this manuscript uses fMRI together with population receptive field (pRF) modeling to investigate the spatial tuning properties of different visual cortical regions (V1, V2, V3, V3ab, hV4, LO) when recalling visual stimuli from memory, compared with perceiving the same stimuli. The central claims of the paper are that (i) memory responses in all regions accurately reflect the spatial location of the recalled stimuli during the memory task; (ii) memory responses are less precise than perception responses; (iii) while perception responses show a precision gradient from high to low along the forward visual hierarchy, this gradient is absent, and even tends to be reversed, for memory responses; (iv) this reversal cannot be explained by lower signal-to-noise ratio during memory trials; (v) instead it is well explained by a hierarchical cortical model assuming a feedback flow of information from high- to low-level cortical areas. These claims are based on solid evidence from a simple visuo-spatial memory task, and straightforward modeling of the empirical and simulated data. There are some obvious shortcomings, like the small sample size ($n=9$) and the fact that the empirical data, if anything, only show a weak trend towards reversed precision, which the authors fully acknowledge in the manuscript. Despite these weaknesses, the results are generally convincing. The paper adds interesting novel support to the notion that memory engages similar neural resources as perception, but with a mainly feedback hierarchical information flow, a topic of much recent interest in the cognitive neuroscience community.

Major comments:

- (1) Conceptually very similar findings were recently reported by Breedlove et al. (2020, *Current Biology*), in a study using a generative feedback model that predicted fMRI responses to imagined and perceived images very well. It should be clearly discussed how the present manuscript overlaps and potentially goes beyond this work.
- (2) Throughout the manuscript, parameters are derived from von Mises distributions that are fit to the average perception and memory responses across subjects, and bootstrapping procedures are then used for statistical testing. It would be reassuring to see (e.g. in a supplemental figure) how well the von Mises distributions and their parameters fit the individual participant data, especially for the memory data. Showing individual subject data is important given the small sample size of this study ($n=9$).
- (3) The results section (l. 137-182) reports the main effect of region, and the region-by-task interaction for precision (FWHM), but not for location or amplitude. The interaction for amplitude is interesting because, looking at Fig. 4c, the amplitude difference between perception and memory decreases along the visual hierarchy, mirroring the pattern found for precision (i.e., FWHM). The authors should report statistics for this interaction along with the main effect of task, and discuss the theoretical implications of this mirrored pattern (i.e., what it means, and why the reverse pRF model would predict such a mirrored pattern, potentially indicating a high inverse dependency between precision and amplitude).
- (4) The results of the simulations are described in a largely qualitative manner. Statistical support (e.g., goodness-of-fit indicators) should be provided for the two main conclusions that (i) a feed-forward model cannot capture the central characteristics of the memory responses well, and that (ii) a bidirectional hierarchical model can capture them well. At present, the only statistical evidence regarding (i) comes from the confidence intervals describing the slope of model predictions relative to actual ROI data (paragraph starting line 268), which seems quite indirect; while with respect to (ii), the comparison between model predictions and memory response is purely qualitative.
- (5) The simulations for memory errors (Fig. 5c), and partly also retrieval failures, are counterintuitive

when thinking about what spatial pattern such errors would most likely generate. To simulate memory errors, the authors rotate response data by a random error angle, drawn from a normal distribution (with varying standard deviations to simulate different degrees of errors). To me, however, it seems highly unlikely that participants, when committing an error, would mentally place the stimulus at random locations around the true location. Since only 4 spatial locations are possible in this paradigm, and participants learn these possible locations extremely well during the behavioural training, it seems far more likely that when committing an error, they will mentally place the image at other possible locations (that is, at 0°, 90°, 180° or -90° from the true location). An additional simulation for such categorical errors should be included in the paper. Such a simulation might also capture retrieval failures quite well, because even in the absence of memory on a given trial, participants will not have forgotten the 4 possible goal locations. The way memory failures are simulated in the current manuscript better approaches attentional lapses on some trials, where the response might then really be entirely flat (as indicated in the discussion).

(6) The authors conceptualize their task as an episodic memory task, and in some instances refer to the hippocampus as providing the drive for the reverse propagation during memory recall (e.g., from line 393). Can the authors provide any evidence, from their empirical data, that hippocampus contributes to the precision of memory recall, e.g. by splitting trials into those with high and low precision, and comparing hippocampal activity between them? Relatedly, it seems that the task used here is quite removed from real-world episodic memory in that it uses highly artificial and over-trained stimuli. It would be good to see a brief discussion of how much, in the authors' opinion, the present results can be generalized to real-life episodic (i.e., one-shot contextual) memory.

(7) Can the authors explain to the reader why there is such a strong dichotomy between V1-V3 on the one hand, and hV4-V3ab (see Fig. 5 rightmost panels) when it comes to simulating FWHM by adding noise or memory failure/errors?

Minor:

- A reference should be provided for the statement that V1-hV4 can be sorted hierarchically with the highest confidence (l. 167/168).
- A recent study by Bone et al. (2020, Nature Communications) touches on similar questions (i.e., what features of an image are reactivated during memory recall), and should be cited.
- A brief justification should be added for the selection of ROIs, and the focus on ventral visual regions in this spatial tuning paradigm, especially the higher-level ROIs (hV4, LO, V3ab).

Reviewer #2:

Remarks to the Author:

This is a well written manuscript and technically rigorous study. The authors use fMRI and pRF mapping to assess the amplitude and spatial precision of responses in retinotopic visual cortex for perception and memory. The main finding is that responses decrease in precision when ascending the visual hierarchy for perception, whereas memory (or imagery) leads to broadly tuned responses that are quite similar in width (FWHM) across visual areas. The authors perform a variety of control analyses to make a convincing case, by manipulating the SNR of voxel responses, considering the possibility of memory failure, and a simulation of memory error. These findings support the case that feedback effects of memory lead to a different pattern of spatial tuning than feedforward perception. The authors present a 1D convolutional model to show how feedback might lead to only a modest coarsening of precision at the lowest level of the hierarchy.

The authors partly address the issue of spatial attention in the discussion section but a more rigorous test to show that more is going on here would be to see if activity patterns in early visual areas can distinguish the remembered item. For example, are activity patterns for the remembered radial frequency pattern at a particular location more similar to that same pattern when presented for perception. This can be tested, since all 4 patterns were presented in the perception condition. If the detailed pattern cannot be predicted, then what is being measured could simply reflect a spatial spotlight of attention rather than a representation that includes some visual content.

The other question is whether the current study is of broad enough interest to warrant strong consideration for publication in Nature Communications. The study is technically solid, and adds to the literature on the types of responses one would expect to see in cases of imagery or retrieval from long-term memory, but it overlaps to a considerable degree with the extensive fMRI literature on visual working memory. In this context, the findings appear less novel and are broadly consistent with the studies in that area while providing additional analysis controls to show a similar broad tuning across visual areas in the memory condition.

Reviewer #3:
Remarks to the Author:
General

This is a well-written and well-structured paper, with a clear progression through the data, hypotheses, and models. The authors manage to make sophisticated modeling methods appear straightforward and readily comprehensible. Overall, I only have one major concern, and I'm not even sure this concern would invalidate the final conclusion of the paper – that differences between sensory and memory representations of space emerge due to the direction in which information travels through the visual hierarchy (i.e. bottom up vs top down).

Signed,
Rosanne Rademaker

Major comment

(1) My one major concern is that there is no objective measure of performance during the memory task, making it difficult to assess what subjects are really doing during this task. While the vividness measure might seem sufficient, it may not be due to several reasons. First, as far as I understand, this measure does not specify what is vivid about the memory. Was it the radial frequency pattern? Was it the location? Was it both? For example, some subjects may decide to recall/imagine the radial frequency pattern close to fixation while verbalizing the location ("1.30 on the clock", for example). This subject may then end up rating the vividness of the pattern recall, and have overall high vividness ratings. Even assuming a not-quite-so-annoying subject – high vividness ratings may still result if subjects weigh the vividness of the pattern more than the vividness of its location when they are providing their ratings. Second, the memory task is effectively categorical. Yes, subjects need to retain location with some detail during training, this requirement is lifted in the scanner. Again, let's assume some pesky subject who decides it's easier to systematically recall the pattern closer to fixation than it actually was (i.e. not at 2°), or to imagine a line extending from fixation to the corner of the screen, or imagine the fractal as large as the quadrant in which it was presented. All of these options could lead to the memory responses being quite different from the sensory responses - especially when the authors reduce the analysis to vertices centered on PRF locations near the sensory stimulus eccentricity. What helps is that 2 of the authors were subjects (giving at least an introspective account of these subject's strategies), and the hierarchical model matches trends in the data quite well. Yet, I remain concerned about the overall poor experimental control over what subjects might be doing in the memory task.

Minor comments:

- (1) Related to the above comment of the memory task being effectively categorical, I wanted to follow up on modeling the memory errors. The authors say they have to assume implausibly large errors (i.e. 45°) for this model to approach the memory data. However, what if subjects accidentally mix up two cues, and thus confuse two quadrants with each other. Such a misbinding of cue-to-quadrant would lead not to small errors, but to errors of any multitude of 45° .
- (2) Line 74: For interpretation of results going forward, it would be useful to include #subjects here.
- (3) Figure 4a: For clarification, I assume the left two panels are a single subject, while the right most panel is the average across all subjects, correct?
- (4) Figure 4: While it's clear from the data that dark-orange is probably perception (and light orange is memory), a legend would be useful for the lazy reader. Especially since the figure legend does not clarify either – some potential for confusion.
- (5) Line 146: "the fact the" should probably be "the fact that"?
- (6) Are amplitude and FWHM truly independent in this fitting procedure? In the absence of a good intuition, is it possible to see some kind of scatter plot of the two plotted against each other with dots for each bootstrap.
- (7) line 200: how is empirical SNR calculated?
- (8) line 239-140: typo? Or is the claim really that subjects experiencing a small, variable amount of memory error explains the memory data?
- (9) I am kind of surprised at the shape of the V1-3 lines in Fig 5b (right) for the FWHM plot. It seems that at both 0% and 100% retrieval failure, the proportion of simulations that fall within the memory CI should be 0. This because at 100% retrieval failure the 1D polar angle response function is presumably flat, and FWHM approaches inf, i.e. always $>$ the memory CI. I also would have expected that there would be some sweet spot in between 0% and 100% failure where the proportion within the memory CI would be quite high. Probably, I just have a wrong intuition about the analysis, but why is this not an inverted U shape for V1-3? Supp fig 1b does not convince me that 100% failure curves are not flat.
- (10) Line 312 (and 406-409): Would it help to test this by collapsing early and later visual ROIs (i.e. "giant" early V1-3 ROI vs. "giant" late ROI hV4/LO/V3ab)? This might show a significant reversal.
- (11) With respect to the hierarchical model, how were kernel sizes chosen? did the authors try to fit the size of the convolutional kernel to the data? For example, if fit only to the perception data, how well can memory data be predicted from kernel sizes derived from fitting perception data? And when both perception and memory data are used to constrain the model?
- (12) Out of curiosity, did the authors explore the relationship between vividness ratings and 1D response functions?

We thank the reviewers for their many helpful suggestions. We have thoroughly addressed all comments through new analyses, figures, and edits to the manuscript. We believe the revised manuscript is substantially improved.

Reviewer #1 (Remarks to the Author):

The study reported in this manuscript uses fMRI together with population receptive field (pRF) modeling to investigate the spatial tuning properties of different visual cortical regions (V1, V2, V3, V3ab, hV4, LO) when recalling visual stimuli from memory, compared with perceiving the same stimuli. The central claims of the paper are that (i) memory responses in all regions accurately reflect the spatial location of the recalled stimuli during the memory task; (ii) memory responses are less precise than perception responses; (iii) while perception responses show a precision gradient from high to low along the forward visual hierarchy, this gradient is absent, and even tends to be reversed, for memory responses; (iv) this reversal cannot be explained by lower signal-to-noise ratio during memory trials; (v) instead it is well explained by a hierarchical cortical model assuming a feedback flow of information from high- to low-level cortical areas.

These claims are based on solid evidence from a simple visuo-spatial memory task, and straightforward modeling of the empirical and simulated data. There are some obvious shortcomings, like the small sample size (n=9) and the fact that the empirical data, if anything, only show a weak trend towards reversed precision, which the authors fully acknowledge in the manuscript. Despite these weaknesses, the results are generally convincing. The paper adds interesting novel support to the notion that memory engages similar neural resources as perception, but with a mainly feedback hierarchical information flow, a topic of much recent interest in the cognitive neuroscience community.

Major comments:

(1) Conceptually very similar findings were recently reported by Breedlove et al. (2020, *Current Biology*), in a study using a generative feedback model that predicted fMRI responses to imagined and perceived images very well. It should be clearly discussed how the present manuscript overlaps and potentially goes beyond this work.

Thank you for bringing up this interesting recent paper. The findings by Breedlove et al. (2020) are indeed relevant. A discussion of Breedlove et al. was not included in our original manuscript because the paper was only published after our manuscript was under review. Because the two sets of studies were conducted independently, it should be clear that our paper is not a response to theirs, but that it is a study conducted at the same time which reaches related conclusions. We read their paper carefully and comment on some of the similarities and differences below. We incorporate the most important of these issues into our manuscript, as noted below.

Overlap. As the reviewer notes, Breedlove et al. report some conceptually similar findings. Both our paper and Breedlove et al. reach the conclusion that perceptual and mnemonic activity are different in human visual cortex. We believe that these are the first two papers to clearly and explicitly demonstrate this. The fact that we both reach this conclusion using different stimuli and methods strengthens the conclusions of both papers, and helps establish this important point. Both papers show that one of the ways that the mnemonic response differs from the stimulus-driven response is that the mnemonic response shows coarser spatial representations in early visual areas. We now explicitly note this in the Discussion.

“Other recent work has also argued for reconsidering a strict view of reactivation. A recently published study using similar methods reported a highly complementary finding to ours: that individual voxels in human visual cortex have larger pRFs during imagery than during perception (Breedlove et al., 2020).” (Discussion, lines 526-528)

Major Differences. Our papers also have important differences, which we detail below.

Model predictions. Our hierarchical model makes the surprising prediction that population responses should be coarser in early visual areas than late visual areas during memory, the opposite of what it predicts during perception. This prediction is not made in the Breedlove et al. model. While their model predicts that responses should be coarser during imagery than perception, they nonetheless predict that the earlier areas should show smaller RFs than later areas during imagery (see Fig 1c in their paper). Thus, their model predicts the *same* pattern across visual areas during both perception and memory while our model predicts *opposite* patterns. We believe that this difference is conceptually significant, and that a key contribution of our paper is that we propose a theoretically-guided and anatomically plausible framework for a reversal of the perceptual pattern during memory retrieval. We make this contrast explicit in the Discussion and comment on its implications for future work in this area.

“Our model predictions diverge from those of a recently published generative model of visual imagery (Breedlove et al., 2020) in predicting this reversal. Like our model, the Breedlove et al. model predicts overall coarser spatial responses during imagery than during perception. However, whereas the Breedlove et al. model predicts that late visual areas should have coarser spatial responses than early visual areas during both perception and memory, we predict a reversal of the perception pattern during memory. Future work should adjudicate between the predictions of these models with a sufficiently powered experiment.” (Discussion, lines 568-573)

Ruling out confounds. The Breedlove et al. paper is primarily a modeling paper with some support from 3 fMRI subjects. The biggest contribution is their generative model. Our paper is primarily empirical, with some support from a hierarchical model. As such, our model is much simpler than theirs. However, we go into greater depth in analyzing certain aspects of the functional imaging data than they do. In particular, we demonstrate with a large number of simulations and analyses that the differences between mnemonic responses and perceptual

responses are not explained by confounds. Though Breedlove et al. show that lower SNR during imagery cannot explain the changes in spatial frequency tuning they observe (Fig S4 in their paper), they do not show the same for the changes in receptive field size and they do not model the effects of specific types of performance failures during memory. We show, with direct simulation, that reduced SNR during memory cannot explain the wider spatial tuning we observe in V1-V3 during memory than perception. Additionally, we now simulate three different kinds of cognitive failures in the memory task (task lapses, angular error, and associative errors; for the associative error simulation see our response to point 5) and show that *none* of these failures can explain our data.

In our view, these additional analyses in our manuscript are not minor details. A dominant theory is that memory responses differ from perceptual responses primarily with respect to signal amplitude and fidelity. Convincing demonstrations that lower SNR and memory failure cannot explain our memory results is key to supporting the central claim in our paper. We now highlight the fact that our analyses establish these facts beyond what's reported by Breedlove et al. in the Discussion (see next point for text).

Voxel-level vs population-level results. Breedlove et al. report differences in individual voxel receptive field sizes between perception and imagery, while we report differences in the spatial tuning of the population response across many voxels. Though these may be related in principle, it is difficult to determine whether our results and theirs are in precise agreement because our results are evaluated at a specific eccentricity and Breedlove et al.'s voxel results are collapsed across all eccentricities. Differences in voxel-wise and population analyses have been previously reported (Sprague and Serences, 2013), and in fact we report one such difference in our paper (line 357-358, Fig. 6). We point out this difference and the prior one in the Discussion.

“While our empirical results and those of Breedlove et al. 2020 are mutually supportive, our results are also distinct from theirs because we quantify neural responses at the population-level rather than voxel-level and rule out numerous confounds related to memory task performance.” (Discussion, lines 528-531)

Minor Differences. There are other more minor differences between our paper and Breedlove et al. (2020).

Behavioral training. We evaluate subjects' memory for both the stimulus pattern and spatial position prior to scanning and require subjects to reach a behavioral criterion in order to participate in the scan session. Breedlove et al. allows subjects to determine for themselves when they have learned the images in the imagery task (see Methods pg e2 in their paper). We believe the fact that we formally evaluated subjects' learning prior to making neural measurements strengthens confidence in our results.

Scholarship and reference lists. While our paper frames itself mostly within the context of episodic memory reactivation, we also cite many mental imagery and visual working memory papers. Breedlove et al. cites mental imagery papers but their paper is also framed strongly in the context of generative models and cites many artificial intelligence papers. Our reference lists overlap very little. We point this out merely to say that we think our papers make different connections with nearby fields and are likely to reach different audiences.

(2) Throughout the manuscript, parameters are derived from von Mises distributions that are fit to the average perception and memory responses across subjects, and bootstrapping procedures are then used for statistical testing. It would be reassuring to see (e.g. in a supplemental figure) how well the von Mises distributions and their parameters fit the individual participant data, especially for the memory data. Showing individual subject data is important given the small sample size of this study (n=9).

We agree that visualizing and evaluating data from individual subjects is important.

To address this, we have added a new section to the Results that reports the reliability of our results in individual subjects: “Differences between perception and memory responses are not explained by inter-subject variability” (beginning line 235). We have also added a figure (see below) that plots the perception and memory response curves in each ROI for each subject (Supplementary Fig. 2a,b). Visually, there is good reliability across subjects during both perception and memory and no indication that one or two subjects may be driving the group average difference we report between perception and memory. To report individual subject data quantitatively, we now also plot location, amplitude, and FWHM parameters for each subject (Supplementary Fig. 2c). The relationships we report in the group average data are clearly evident in the majority of individual subjects. Finally, we also recompute the critical statistics using individual subject estimates (paragraph beginning line 252). These tests support the conclusions reached from the bootstrapped confidence intervals.

Supplementary Figure 2

(3) The results section (l. 137-182) reports the main effect of region, and the region-by-task interaction for precision (FWHM), but not for location or amplitude. The interaction for amplitude is interesting because, looking at Fig. 4c, the amplitude difference between perception and memory decreases along the visual hierarchy, mirroring the pattern found for precision (i.e., FWHM). The authors should report statistics for this interaction along with the main effect of task, and discuss the theoretical implications of this mirrored pattern (i.e., what it means, and why the reverse pRF model would predict such a mirrored pattern, potentially indicating a high inverse dependency between precision and amplitude).

Thank you for highlighting this point. We now report interactions for location and amplitude in the text of the Results.

Location (Results, lines 154-156):

“Similarly, there was no overall difference between ROIs (main effect of ROI: $\beta = 0.075$, 95% CI = [-2.22, 2.65]) and no change in the relationship between perception and memory across ROIs (perception/memory x ROI interaction: $\beta = 0.54$, 95% CI = [-1.94, 2.60]).”

Amplitude (Results, lines 179-185):

“During both perception and memory, we observed lower response amplitudes in later areas (main effect of ROI: $\beta = -0.023$, 95% CI = [-0.035, -0.008]; Fig. 4c, middle). Interestingly, however, there was a larger decrease in amplitude from early to late areas during perception than during memory (perception/memory x ROI interaction: $\beta = -0.12$, 95% CI = [-0.16 -0.082]). This resulted in perception and memory amplitudes that were more similar to each other in the later areas than in the earlier areas. For example, while perception amplitudes were 0.86 units of % signal change larger than memory amplitudes in V1, they were only 0.37 units of % signal change larger in V3ab.”

As noted in the quoted text above, we agree with the Reviewer that the amplitude interaction is interesting and worth noting. It is consistent with the FWHM interaction in that the difference between perception and memory is smallest in the latest visual areas and maximal in V1. From the perspective taken by our hierarchical model, both of these interactions are caused by the fact that there are fewer pooling transformations between V3ab responses during perception and V3ab responses during memory than there are between V1 responses during perception and V1 responses during memory. However, we note that the main effect of ROI on amplitude in our data is difficult to interpret because it is likely dependent on the stimulus used. Our hierarchical model does not consider stimulus selectivity apart from selectivity for space, but non-spatial factors have a large influence on the amplitude of the BOLD response. Our stimuli may be better targeted to the feature preferences of some areas more than others and the use of a different stimulus set (e.g. faces) in the same visual field locations could alter the amplitudes we observe considerably. It is also easy to think of ways to break the inverse relationship between amplitude and FWHM that the reviewer mentions by using a different stimulus. For instance, presenting a high contrast ring stimulus would yield a high amplitude and very wide response in V1, while presenting a low-contrast but punctate stimulus would yield a low amplitude but narrow response in V1 (see also our response to Reviewer 3, point 6). Thus, while we agree with the Reviewer that the interaction is interesting, we are cautious about interpreting the absolute value of the amplitudes and the main effect of ROI.

(4) The results of the simulations are described in a largely qualitative manner. Statistical support (e.g., goodness-of-fit indicators) should be provided for the two main conclusions that (i) a feed-forward model cannot capture the central characteristics of the memory responses well, and that (ii) a bidirectional hierarchical model can capture them well. At present, the only statistical evidence regarding (i) comes from the confidence intervals describing the slope of model predictions relative to actual ROI data (paragraph starting line 268), which seems quite indirect; while with respect to (ii), the comparison between model predictions and memory response is purely qualitative.

We respond to each of these points separately:

(i) A feed-forward model cannot capture the central characteristics of the memory responses well.

We now provide quantitative support for this claim in three figures. As the reviewer notes, the first is Figure 6c, which plots the predicted amplitude and FWHM from the pRF model against the actual amplitude and FWHM during perception and memory. A major result of our paper is that changes in FWHM across ROIs are different for perception and memory (Fig. 4c). The analysis provided in Figure 6c (including the estimated slopes and confidence intervals) directly assesses how well the pRF model can predict these across-ROI patterns. The differences between the memory plots and the perception plots are not subtle. The perception data are close to the model predictions, whereas the memory data is nearly orthogonal for FWHM. We believe these plots with bootstrapped confidence intervals are quite convincing in showing that the stimulus-driven pRF models fail to capture important features of the memory data. For the sake of thoroughness, we have added two more analyses. In one, we compare perception and memory FWHM and amplitude to predictions from two simpler pRF models (Supplementary Fig. 5). These analyses show the same failures for the memory data, and demonstrate that our results are not specific to the particular pRF model used. Second, rather than assessing the accuracy of the pRF model in predicting properties of interest derived from the tuning functions (amplitude and width), we directly compute the goodness of fit (R^2) between the tuning function predicted from the pRF model and the observed tuning function (Supplementary Fig. 6). Because the amplitude of the memory data is much lower than predicted by the pRF model, the pRF model fails catastrophically to predict the memory data (Supplementary Fig. 6, upper panel). Even if we allow an arbitrary re-scaling of the pRF model predictions, they are less accurate at predicting the memory data than the perception data (Supplementary Fig. 6, lower panel).

We explicitly discuss the R^2 analyses in the Results (paragraph starting line 378):

“In addition to assessing how well our pRF model captured across-ROI changes in FWHM and amplitude, we directly quantified the goodness-of-fit between the pRF model predictions and our experimental data in each ROI. To do this, we computed the R^2 between the predicted and observed polar angle response functions, separately for each ROI and task. Goodness-of-fit was very high for the perception data in every ROI (range: 0.65--0.93). In contrast, goodness-of-fit was extremely poor for the memory data. R^2 values were negative in every ROI during memory (range: -11.9 -- -1.84), indicating that the pRF model failed to capture the mean response (Supplementary Fig. 6, top). This is largely driven by the fact that the mean predicted amplitude from the pRF model was much higher than the mean amplitude of the memory data. We wondered to what extent this failure could be corrected by rescaling the pRF predictions to be lower in amplitude. After rescaling the pRF predictions with the single scale factor that best approximated the perception data or the memory data, we re-evaluated R^2 . While rescaling improved R^2 for

the memory data in every ROI, R^2 for the memory data still fell below R^2 for the perception data in most ROIs, with the largest failure in V1 (Supplementary Fig. 6, bottom). These analyses demonstrate that our pRF model is a good predictor of the perceptual responses we measured, but a poor predictor of the mnemonic responses we measured.”

We also now highlight that the analyses we present in Figure 5 generalize to other pRF models:

“In order to rule out the possibility that this finding is specific to the particular pRF model we used, we performed the same analysis on two simpler, previously published, pRF models (a CSS model and a linear model; see Methods). We find that these models are also dramatically better at capturing across-ROI patterns in FWHM and amplitude for the perception data than for the memory data (Supplementary Fig. 5b).” (Results, lines 373-377)

(ii) a bidirectional hierarchical model can capture them well

We are hesitant to quantify the goodness of fit between our data and the hierarchical model since we intended the hierarchical model to be qualitative. No attempt was made to fit model parameters to the data or to produce model predictions that exactly estimate FWHM values. A much more complex model that accounted for additional variables such as stimulus selectivity and that attempted to reproduce our data could, in principle, be developed, but that is beyond the scope of this manuscript. Instead, our goal was to understand which assumptions and computations are required to reproduce the large trends across region and task, regardless of the exact model parameters used. We now explain our logic more explicitly in the Results:

“Here, we explored whether a simple hierarchical model with spatial pooling could capture the qualitative pattern of results we observed. We were specifically interested in whether a single model could account for changing patterns of precision and amplitude between perception and memory retrieval. We asked whether manipulating the direction of information flow within the model would be sufficient to account for these qualitative differences without changing any components of the model structure or parameter values.

Importantly, we did not fit our model to our data. Nor did we attempt to generate a model that would predict the exact FWHM and amplitude values we report. Instead, we sought to identify a *class* of models where most parameter choices yielded the qualitative pattern of data we observed during both perception and memory.” (Results, lines 406-414)

A key part of our argument is that our model can capture basic trends in our data across a wide range of parameter values. To better support this, we have added a figure (see below) that plots model FWHM across a range of parameter values (Supplementary Fig. 7). This figure demonstrates that our model captures qualitative data trends among models with different numbers of layers (Supplementary Fig. 7a), and with different stimulus and kernel sizes (Supplementary Fig. 7b). We highlight these plots explicitly in the Results:

“We note that while the exact FWHM values depend on the number of layers, the stimulus size, and the kernel sizes (Supplementary Fig. 7), the qualitative patterns we focus on are general properties of this model class and can be observed across many parameter choices. Though some parameter choices can result in little to no change in FWHM, however, there are no parameter choices that can reverse the trends we report -- for example, by producing *smaller* FWHM in layer 1 than in layer 2 during feedback.” (Results, lines 444-448)

Importantly, we think that capturing these qualitative trends is more important than capturing the exact FWHM values we report, which may vary according to the stimulus and task.

Supplementary Figure 7

(5) The simulations for memory errors (Fig. 5c), and partly also retrieval failures, are counterintuitive when thinking about what spatial pattern such errors would most likely generate. To simulate memory errors, the authors rotate response data by a random error angle, drawn from a normal distribution (with varying standard deviations to simulate different degrees of errors). To me, however, it seems highly unlikely that participants, when committing an error, would mentally place the stimulus at random locations around the true location. Since only 4 spatial locations are possible in this paradigm, and participants learn these possible locations extremely well during the behavioural training, it seems far more likely that when committing an error, they will mentally place the image at other possible locations (that is, at 0°, 90°, 180° or -90° from the true location). An additional simulation for such categorical errors should be included in the paper. Such a simulation might also capture retrieval failures quite well, because even in the absence of memory on a given trial, participants will not have forgotten the 4 possible goal locations. The way memory failures are simulated in the current manuscript better

approaches attentional lapses on some trials, where the response might then really be entirely flat (as indicated in the discussion).

Thank you for suggesting this analysis, which was also suggested by Reviewer 3. We now include an analysis that simulates associative memory errors as the reviewer describes. Similar to our other simulations, very high error rates (>50% associative errors) are required to artifactually produce something similar to the memory responses we observed with no underlying changes to the neural response properties. Given that subjects made no associative errors in the last round of training, we find this error rate to be implausible. Further, like the other simulations, simulating a large number of associative errors captures our memory data well in V1, but very poorly in higher areas (see also our response to point 7 for clarification on this issue). We thank the reviewer again for suggesting this analysis, as we believe it strengthens the conclusions of the paper.

We report the results of this simulation in the Results: “Associative Error Simulation” (starting line 319). We now plot the results in Figure 5c (see below), Supplementary Figure 3c, and Supplementary Figure 4c. We also agree with the Reviewer that the “retrieval failure” simulation is better conceptualized as a task lapse, and we adopt this new terminology in our revisions.

Figure 5c

(6) The authors conceptualize their task as an episodic memory task, and in some instances refer to the hippocampus as providing the drive for the reverse propagation during memory recall (e.g., from line 393). Can the authors provide any evidence, from their empirical data, that hippocampus contributes to the precision of memory recall, e.g. by splitting trials into those with high and low precision, and comparing hippocampal activity between them? Relatedly, it seems that the task used here is quite removed from real-world episodic memory in that it uses highly artificial and over-trained stimuli. It would be good to see a brief discussion of how much, in the authors’ opinion, the present results can be generalized to real-life episodic (i.e., one-shot contextual) memory.

We thank the reviewer for raising this point. Indeed, our theoretical framework centers around computational models that specify hippocampal pattern completion as critical to driving cortical

reactivation. We chose a paired associates task for this experiment because of its history of use in studying hippocampally dependent memory (Parkinson, Murray, & Mishkin, 1988; Gilbert & Kesner, 2003) and cortical reactivation (Nyberg et al., 2000; Kuhl et al., 2011). Based on these (and other) data points which have borne out the basic predictions of this framework, we believe it is reasonable to reference it. We agree with the reviewer that this task varies from real-life memory. As the reviewer notes, real-life experiences take place only once and have many complex features, only some of which are remembered. In this experiment, we intentionally avoided these aspects of human memory. There is already good reason to believe that memory reactivation differs from perceptual activity due to selective or failed encoding, forgetting, memory distortion, etc. In this case, we wanted to test whether memory reactivation has distinct properties from perceptual activation *even when these factors are not present*, i.e. when the memory system is performing optimally. Thus, we relied on a highly simplified paradigm with a very low memory load and extensive training. We have briefly clarified our reasoning on this point in the Introduction:

“Critically, any systematic differences between visual cortical response properties during perception and memory should be observable under conditions of high memory fidelity, when memory strength and accuracy are maximized.” (Introduction, lines 55-57)

Now that we have established this, we think extending this line of work to examine responses in single-shot memory paradigms is an important next step, and we have already collected data for such a follow-up experiment. We agree with the reviewer that probing the relationship between hippocampal activity and the precision of spatial activity in visual cortex is of high interest. We think this would be most appropriate to pursue in a paradigm where there is variability in subjects’ neural and behavioral memory precision. As previously stated, we are actively working on this. In the present experiment, however, we expect most of the trial-to-trial variability in neural precision to reflect measurement noise rather than variability of interest.

(7) Can the authors explain to the reader why there is such a strong dichotomy between V1-V3 on the one hand, and hV4-V3ab (see Fig. 5 rightmost panels) when it comes to simulating FWHM by adding noise or memory failure/errors?

We agree that the intuition for this pattern was not sufficiently clear in our original manuscript. The critical difference is that while V1-V3 FWHM values are much smaller during perception than during memory in our data, they are equivalent in hV4-V3ab. Thus, when we apply noise to the perception responses, effectively blurring them, this makes V1-V3 responses *more similar* to the memory responses (higher values on the y axis in Fig 5), but it makes hV4-V3ab responses *less similar* to the memory responses because they were equivalent to start (lower values on the y axis in Fig 5). Said another way, the strong dichotomy the reviewer refers to reflects the fact that the empirical relationship between perception and memory responses is categorically different for V1-V3 and hV4-V3ab, and the blurring imposed by the simulations thus has different effects in these areas. We clarify this intuition in the Results:

“First, at high levels of noise (low levels of SNR), any modest increase in ability to capture V1-V3 FWHM was accompanied by a *decrease* in ability to capture FWHM in later visual areas (Fig. 5a, right, and Supplementary Fig. 3a and Supplementary Fig. 4a). In these later regions, FWHM was empirically equivalent during perception and memory, and artificially adding noise to the perception data eliminates this equivalence.” (Results, lines 303-307)

However, we have realized our original visualization of these results did not make it clear as to whether the simulated FWHM values do not match the memory data because they are too small or too large, which is critical for interpreting the results. Thus, we now additionally plot the von Mises fits to simulated data in two different ROIs (Supplementary Fig 3) and plot the simulated FWHMs with respect to the actual memory FWHM values (Supplementary Fig 4, see below for example). In these plots, one can see that simulations from the no noise condition (leftmost xtick) fall below the memory confidence intervals (shading) in V1-V3 and within them from hV4-V3ab. As noise is added, FWHM increases in all ROIs. However, this causes more simulations to fall within the memory confidence intervals in V1-V3 and fewer in hV4-V3ab. We believe these additional plots will make the results of the simulations more clear to readers, and we thank the reviewer for raising the point.

Supplementary Figure 4d

Minor:

- A reference should be provided for the statement that V1-hV4 can be sorted hierarchically with the highest confidence (l. 167/168).

Thank you for this comment. We’ve removed this statement from the Results section (and the accompanying statistic) since no strong conclusions could be drawn from analyzing this subset of the ROIs. That said, we note that the hierarchical relationship between V1-V3 is supported by anatomical tracing studies (Felleman and van Essen, 1991; Markov et al., 2013). It is somewhat less clear how to consider V4, LO, and V3ab with respect to each other. Part of this difficulty arises from the lack of clear homology between these areas and those in the macaque (Tootell, 2001; van Essen et al., 2001).

- A recent study by Bone et al. (2020, Nature Communications) touches on similar questions (i.e., what features of an image are reactivated during memory recall), and should be cited.

Thank you for alerting us to this interesting recent paper. We agree it is relevant, and we now cite this paper in the Discussion with regards to the use of encoding models to study memory reactivation (line 485) and with regards to the finding of reactivation in non-sensory areas (line 584).

- A brief justification should be added for the selection of ROIs, and the focus on ventral visual regions in this spatial tuning paradigm, especially the higher-level ROIs (hV4, LO, V3ab).

Our goal in selecting ROIs was to select regions that could reliably be defined according to retinotopic maps and that varied in their placement in the visual hierarchy. We selected V1, V2, and V3 because they are well-studied early visual regions that are very easy to define retinotopically and that have a hierarchical structure that most agree on (eg. V2 receives most inputs from V1). We then selected one higher-level ROI bordering V3 from each of the ventral, lateral, and dorsal streams: hV4, LO, and V3ab, respectively. We don't think this sample is biased toward the ventral stream, as it includes one lateral stream ROI and one dorsal stream ROI. It's true that some have argued that dorsal stream areas may be particularly important for spatial processing (Mishkin et al., 1983). In our experience, more anterior dorsal maps in the IPS are much harder to define retinotopically than the ones we selected; they are smaller and noisier and thus require more data or specialized pRF mapping tasks (Mackey et al., 2017). Future experiments will be needed to assess responses in these areas.

We briefly highlight our ROI selection in the Results:

“We generated these polar angle response functions for V1--V3 and for three mid-level visual areas from the ventral, lateral, and dorsal streams: hV4, LO, and V3ab (Fig. 4b).” (Results, lines 134-135)

We now also give a more detailed explanation in the Methods:

“We had several goals in mind when choosing these ROIs. First, we wanted ROIs that spanned the visual hierarchy and that were known to have different receptive field sizes. Second, we wanted to avoid the most anterior maps, which require large quantities of data to define reliably in individual subjects. We selected early visual areas V1-V3, based on their well-described anatomical boundaries and organization in humans (Wandell et al., 2007). We then selected the visual areas bordering V3 from each of the ventral, lateral, and dorsal streams.” (Methods, lines 811-816)

Reviewer #2 (Remarks to the Author):

This is a well written manuscript and technically rigorous study. The authors use fMRI and pRF mapping to assess the amplitude and spatial precision of responses in retinotopic visual cortex for perception and memory. The main finding is that responses decrease in precision when ascending the visual hierarchy for perception, whereas memory (or imagery) leads to broadly tuned responses that are quite similar in width (FWHM) across visual areas. The authors perform a variety of control analyses to make a convincing case, by manipulating the SNR of voxel responses, considering the possibility of memory failure, and a simulation of memory error. These findings support the case that feedback effects of memory lead to a different pattern of spatial tuning than feedforward perception. The authors present a 1D convolutional model to show how feedback might lead to only a modest coarsening of precision at the lowest level of the hierarchy.

(1) The authors partly address the issue of spatial attention in the discussion section but a more rigorous test to show that more is going on here would be to see if activity patterns in early visual areas can distinguish the remembered item. For example, are activity patterns for the remembered radial frequency pattern at a particular location more similar to that same pattern when presented for perception. This can be tested, since all 4 patterns were presented in the perception condition. If the detailed pattern cannot be predicted, then what is being measured could simply reflect a spatial spotlight of attention rather than a representation that includes some visual content.

We thank the reviewer for raising this point and giving us an opportunity to clarify our thoughts on the relationship between spatial attention and memory. We provided a detailed discussion of these topics below.

Are subjects reactivating the stimulus patterns in this task?

Analytical constraints. In order to rule out the possibility that our mnemonic effects are just spatial attention, the reviewer suggests testing whether the remembered stimulus pattern is reflected in visual cortical activity. The exact analysis suggested by the reviewer would not selectively capture reactivation of stimulus pattern because stimulus pattern and location are correlated within subjects. Subjects in our experiment saw each stimulus pattern in only one location. Though the analysis the reviewer suggests would “work”, it would also reflect spatial reactivation. In order to disentangle spatial location from stimulus pattern, we would need to present multiple patterns in the same location. When designing the experiment, we realized our design would preclude the analysis the reviewer suggests, but planned to focus our analyses exclusively on space because: 1) the field has verified encoding models of space, but not of shape/pattern; 2) space is coded coarsely on the cortex, so we can get very robust measurements of it with fMRI.

We now clarify our reasons for prioritizing space in the design in the Discussion:

“In contrast to decoding or pattern similarity approaches, encoding models predict the activity evoked in single voxels in response to sensory or cognitive manipulations using a set of explicit mathematical operations (Naselaris et al., 2011). Spatial encoding models have proved particularly powerful because space is coded in the human brain at a scale that is well-matched to the millimeter sampling resolution of fMRI (Engel et al., 1994; Sereno et al., 1995; Dougherty et al., 2003).” (Discussion, lines 479-483)

We also talk about how to answer similar questions about non-spatial dimensions in the Discussion:

“A related question concerns whether pattern tuning differs between perception and memory in a similar manner to spatial location tuning. Answering this question will require developing well-validated encoding models for stimulus pattern that do not currently exist. There has, however, been new progress in this direction. A recent paper by Breedlove et al. (2020) was the first to demonstrate that tuning for a non-location feature (spatial frequency) differs between perception and memory. Developing visual encoding models that can account for increasingly complex stimulus properties will be a critical component in the effort to understand memory representations.” (Discussion, lines 501-507)

Prior literature and training procedure. Additionally, we note that many papers have already shown that non-spatial visual features are represented in visual cortex during memory retrieval. Indeed, orientation (Bosch et al., 2014; Naselaris et al., 2015), shape (Stokes et al., 2009; Kok et al., 2018), pattern (Hindy et al., 2016), and spatial frequency (Naselaris et al., 2015) have all been reported, and we don’t see this as an open question in the literature. Thus, we assume that the reviewer is concerned about whether these signals are present *in this specific task*. Though we planned to focus our fMRI analyses exclusively on space, we took care to use a paired associates paradigm commonly used in episodic memory, where a cue is arbitrarily associated with a multifeatural item. Based on our specific instructions and this design, we see no reason to suspect that subjects are exclusively recalling spatial information. Subjects were trained on both spatial and pattern dimensions and their ability to recall both dimensions accurately was verified prior to the scan. Thus, by the end of training all subjects could easily and accurately recall both the pattern and stimulus location (Fig 1d,e). During memory scans, we instructed subjects to “use the cue to recall the associated pattern in its associated spatial position”. Subjects were not told that our fMRI analyses would focus on space and thus they had no reason to prioritize this dimension during recall. We now clarify these features of the task and the prior literature in the Discussion:

“Because encoding models of space confer such analytical advantages, we specifically designed our fMRI analyses to assess neural coding of spatial location during memory retrieval. The spatial tuning evident in our results and our simulations are consistent with the idea that subjects were in fact remembering the stimulus location during the fMRI

scan. Although spatial location was the focus of our analysis, this does not imply that subjects only attended this dimension of the stimuli. Subjects were not told that fMRI analyses would focus on the spatial location of the stimulus and were trained to remember both spatial location and pattern features of the stimuli. Hence it is likely that subjects were retrieving both location and pattern. Our design does not allow us to demonstrate stimulus pattern reactivation because subjects only saw one stimulus pattern per location. However, prior studies have shown that stimulus dimensions including orientation (Bosch et al., 2014; Naselaris et al., 2015), spatial frequency (Naselaris et al., 2015), shape (Stokes et al., 2009; Kok & Turk-Browne, 2018), and pattern (Hindy et al., 2016) are encoded in visual cortex activity during memory retrieval.” (Discussion, lines 489-499)

Is it critical to show reactivation of pattern information to support our conclusions?

For the reasons outlined above, we think it is likely that subjects are bringing to mind the stimulus pattern during the memory task. However, we disagree with the reviewer that whether or not subjects are doing this is essential for interpreting our results, and that the results of the proposed analysis would dissociate attention from memory.

Additional explanatory power of attention. First, we do not think that reconceiving of our findings as being related to attention rather than memory explains them. That is, if we reframe this paper as comparing “visually-guided spatial attention” and “memory-guided spatial attention”, we do not think this explains *why* spatial tuning in V1 looks so different in these two conditions. To the best of our knowledge, there is no finding that suggests that memory-guided attentional responses in V1 are as precise or less precise than those in V3ab. Thus, the results would still be surprising and novel if called attention. We highlight this important point in the Discussion:

“[A]re the responses we observed during memory retrieval better characterized as long-term memory reactivation or as attention? There are a number of reasons why we don't think it's useful to attribute our findings to attention alone. First, characterizing our results as spatial attention does not explain them. We report that the precision of spatial tuning in memory differs from the precision of spatial tuning during perception. To the best of our knowledge, there is no parallel finding in the spatial attention literature.” (Discussion, lines 629-633)

Importance of space. Second, even if our findings did not generalize to non-spatial dimensions, we still think they would be significant, since spatial location is itself an important feature of perceptual encoding and memory retrieval. Indeed, there is an entire field with many high impact papers that looks solely at spatial reactivation in rodent place cells without looking at non-spatial features. Many would argue that this work is still meaningful even though it concerns only space. We briefly point this out in the Discussion:

“Given these prior demonstrations and given the historical importance of spatial coding for the study of memory (Moser et al., 2015), we believe that our spatial results are important even in the absence of demonstrating pattern reactivation.” (Discussion, lines 499-501)

Separability of memory and attention. Third, suppose we did find evidence for reactivation of stimulus pattern information in the memory task. Would this eliminate the concern that attention is involved? We believe the answer is no. Even with evidence of feature reactivation, one could then ask whether the observed results reflect feature-based attention as opposed to memory. We elaborate on this and on the relationship between attention and memory more broadly in the next point.

How should we think about the relationship between spatial attention and spatial memory?

We would advocate against the idea that memory and attention are completely independent concepts. It is possible, and even reasonable, to think that memory retrieval involves allocating attention to an internally generated representation, as others have already proposed (Chun et al., 2011). One thing we can be certain of is that our memory task is memory-dependent. The only way to derive the associated spatial location from the cue (a colored dot) is by encoding the arbitrary association between the cue and the associate at the beginning of the experiment. Thus, while our task possibly involves attention, it definitely involves memory, and we think it is reasonable to describe it as such. Whether a similar pattern of results is found during other tasks (working memory, standard visual spatial attention tasks, etc) is an interesting empirical question, but we don't think it undercuts the results we report here. We have clarified our perspective on this issue in the Discussion:

“Second, there are meaningful differences between our task and typical endogenous attention tasks. Most attention tasks have no memory component since the cue explicitly represents the attended location or feature. In contrast, in our task, the correct spatial location or pattern cannot be determined from the fixation cue without having previously encoded the association between them and then successfully recalling it. Thus, our task by necessity involves memory. It may also involve attention, to the extent that items retrieved from memory become the targets of internal attention (Chun et al., 2011). From our perspective, it is plausible that the neural responses we observed during memory retrieval resemble those observed when visual attention is deployed in the absence of a stimulus. Future experiments should address this question directly. Further, modeling efforts should address whether it's possible to develop a model of top-down processing in visual cortex that can account for both memory retrieval and attention, or whether separate computational models are needed.” (Discussion, lines 633-643)

(2) The other question is whether the current study is of broad enough interest to warrant strong consideration for publication in Nature Communications. The study is technically solid, and adds to the literature on the types of responses one would expect to see in cases of imagery or retrieval from long-term memory, but it overlaps to a considerable degree with the

extensive fMRI literature on visual working memory. In this context, the findings appear less novel and are broadly consistent with the studies in that area while providing additional analysis controls to show a similar broad tuning across visual areas in the memory condition.

The reviewer argues that our findings are not novel because similar results have been obtained in the working memory literature. We agree with the reviewer that our manuscript did not sufficiently address overlap with the working memory literature and that clarifying this relationship is critical for understanding the novelty of our results. There are two reasons we think that working memory findings do not reduce the novelty of our results.

First, the working memory literature does not show our main result. We show that the three-fold change in spatial precision from V1 to V3ab during perception (an organizing principle of the primate visual system) is entirely eliminated in long-term memory activity. We don't agree with the reviewer's characterization of this result as a control analysis. That mnemonic response properties are *systematically different* from visually-driven response properties in human visual cortex is our main empirical claim and is reflected in the title of our paper. Importantly, we know of no working memory paper that reaches this conclusion. To the best of our knowledge, working memory studies repeatedly report that working memory representations are shared with perception in visual cortex (Harrison and Tong, 2009; Serences et al., 2009; Rahmati et al., 2017; Rademaker et al., 2019). We are not aware of any paper that quantifies working memory representations in the way we do, but an informal assessment of several recent papers (Sprague et al., 2013, Rahmati et al., 2017, and Rademaker et al., 2019) suggests that stimulus reconstructions made from working memory delay period activity are approximately as precise as reconstructions made from perceptual activity. Thus, we respectfully disagree as a point of fact that our main result has been reported in the visual working memory literature.

We agree with the reviewer that *some* of the results reported in our paper are consistent with working memory literature. For instance, the visual working memory literature demonstrates that visual cortex plays a role in the maintenance of visually encoded spatial information and our results support a similar view of long-term memory retrieval. Indeed, several previous long-term memory papers have already made a similar point (Wheeler et al., 2000; Bosch et al., 2014; Hindy et al., 2016), and we use similar methods to those used in many working memory studies to make it again in our paper. However, the main point of our paper is *not* about the similarity in activity between long-term memory and perception; that point is established.

Second, working memory and long-term memory are not the same cognitive processes. We do not think it can be assumed that findings from one of these domains will generalize to the other. Indeed, the extent to which these processes share neural substrates is still a matter of debate (see Jeneson et al. 2012 for an example). We think there are theoretically-guided reasons to suspect that there may be differences in visual cortical activity in working memory and long-term memory tasks, even if both of these tasks produce mnemonic activity in visual cortex. Ultimately, this will need to be tested empirically by comparing working memory and

long-term memory within the same experiment. We do anticipate our findings to be of interest to working memory researchers and hope that our work inspires such comparisons.

We have edited both the Introduction and Discussion sections of the manuscript with these critical points in mind. In the Introduction, we now note that working memory researchers have relied on similar methods to examine delay-period activity. We also try to pre-empt confusion between what this field has reported and what we show in our manuscript.

“Such approaches have already proved effective in answering some questions about the nature of visual working memory representations. Numerous studies have used encoding models to show that visual cortex encodes stimuli maintained in working memory in a similar format to perception (Harrison and Tong, 2009; Serences et al., 2009; Rahmati et al., 2017; Rademaker et al., 2019). However, these studies have emphasized similarity between working memory and perception and have not reported systematic differences between the two. Moreover, there are good theoretical reasons to suspect that results from working memory studies may not generalize to episodic memory. Namely, while working memory is thought to depend on the *maintenance* of perceptual activity that was evoked seconds ago, episodic memory retrieval requires the total reinstatement of perceptual activity that was evoked minutes, hours, or days ago. These different cognitive operations may impose different constraints on stimulus representations.” (Introductions, lines 68-77)

In the Discussion, we now highlight the novelty of our finding more clearly.

“The field of visual working memory in particular has relied on very similar methods to the ones we use here to investigate the role of visual cortex in memory maintenance (Sprague et al. 2013; Ester et al., 2013). Many such studies have shown that early visual areas contain retinotopically specific signals throughout a delay period (Sprague et al., 2013, Sprague et al., 2014, Rahmati et al., 2017). These studies agree with ours in that they demonstrate a role for visual cortex in representing mnemonic information, and establish that some properties of neural coding during perception are preserved during memory. However, to the best of our knowledge, no working memory study has reported precision differences between perceptual and mnemonic activity akin to what we report. Informal assessment of recent working memory papers suggests that stimulus reconstructions made from working memory delay period activity are approximately as precise as reconstructions made from perceptual activity (Sprague et al., 2013; Rahmati et al., 2017; Rademaker et al., 2019). There are theoretical reasons to expect a difference between working memory and episodic memory representations. During typical working memory tasks, a visual cortical representation that was just evoked must be kept activated. However, during episodic memory tasks such as the paired associates task we use here, there is no recently evoked representation of the stimulus in visual cortex. This representation must be created anew, and in standard models of memory retrieval, this process is initiated by the hippocampus, which sits atop the visual hierarchy. As described by our model, this mechanism for reactivating visual cortex may come with unique costs

relative to working memory maintenance. Ultimately, a direct comparison between working memory and episodic memory in the same experiment will be required to make progress on this question.” (Discussion, paragraph beginning line 603)

Reviewer #3 (Remarks to the Author):

General

This is a well-written and well-structured paper, with a clear progression through the data, hypotheses, and models. The authors manage to make sophisticated modeling methods appear straightforward and readily comprehensible. Overall, I only have one major concern, and I'm not even sure this concern would invalidate the final conclusion of the paper – that differences between sensory and memory representations of space emerge due to the direction in which information travels through the visual hierarchy (i.e. bottom up vs top down).

Signed,

Rosanne Rademaker

Major comment

(1) My one major concern is that there is no objective measure of performance during the memory task, making it difficult to assess what subjects are really doing during this task. While the vividness measure might seem sufficient, it may not be due to several reasons. First, as far as I understand, this measure does not specify what is vivid about the memory. Was it the radial frequency pattern? Was it the location? Was it both? For example, some subjects may decide to recall/imagine the radial frequency pattern close to fixation while verbalizing the location (“1.30 on the clock”, for example). This subject may then end up rating the vividness of the pattern recall, and have overall high vividness ratings. Even assuming a not-quite-so-annoying subject – high vividness ratings may still result if subjects weigh the vividness of the pattern more than the vividness of its location when they are providing their ratings. Second, the memory task is effectively categorical. Yes, subjects need to retain location with some detail during training, this requirement is lifted in the scanner. Again, let's assume some pesky subject who decides it's easier to systematically recall the pattern closer to fixation than it actually was (i.e. not at 2°), or to imagine a line extending from fixation to the corner of the screen, or imagine the fractal as large as the quadrant in which it was presented. All of these options could lead to the memory responses being quite different from the sensory responses - especially when the authors reduce the analysis to vertices centered on PRF locations near the sensory stimulus eccentricity. What helps is that 2 of the authors were subjects (giving at least an introspective account of these subject's strategies), and the hierarchical model matches trends in the data quite well. Yet, I remain concerned about the overall poor experimental control over what subjects might be doing in the memory task.

Thank you for this comment. As the reviewer notes, we did *not* ask subjects to report the remembered locations (or patterns) while being scanned. We did this to avoid confounding the

properties of the remembered stimulus with the motor response used to report these properties (hand or eye movement). We reasoned that it was best to avoid this confound. Given the positive results, one might conduct a follow-up study investigating the relationship between behavioral performance and neural responses during memory. Indeed, we are working on a follow-up experiment where subjects make explicit spatial memory judgments in the scanner. We now briefly explain this logic in the Results section.

“We intentionally avoided asking subjects to report the content of their memories while being scanned. We did this to avoid confounding the properties of the remembered stimulus with the motor response used to report these properties (hand or eye movement).” (Results, lines 109-11)

In addition, while there are lapses in every task, we should emphasize that this was an extremely low load long-term memory task. This was by design, as we were interested in characterizing differences between perceptual and mnemonic activity that were not due to forgetting or interference. There were only four associations, there was no overlap among the associations, and the training was optimized to promote fast and accurate learning. Further, the perception and memory tasks were interleaved in the scanner, so subjects were reminded of the associations before and after each memory run. We are confident that all subjects acquired these memories and could express them by the end of the training session, and that they learned both pattern and location features of each stimulus. As the reviewer notes, it is still possible that subjects chose not to comply with the instructions in the scanner. That said, we can rule out some of the specific possibilities that the reviewer raises.

Remembering the quadrant only: Our results and simulations argue against this account. First, our neural data are reliable in estimating the true location of the stimulus in the visual field. For example, the 95% confidence interval around the true polar angle location of stimuli in memory in V1 is $[-10.25^\circ, 10.54^\circ]$. This indicates that subjects are not just remembering the quadrant that the stimulus is in, but also remembering the angular position of the stimulus within that quadrant. Second, we directly simulate the impact of angular error on neural responses in Figure 5d. While we can't rule out that subjects *never* make angular errors (and we would agree that this would be nice to measure on a trial-by-trial basis), we can conclude that any reasonable error rate does not explain our results.

Also related to the reviewer's question, we have added another simulation to our manuscript to rule out the possibility that associative errors (recalling one of the other 3 studied locations) can explain our results (Figure 5c). In brief, we find that such errors cannot account for our results. See our response to minor comment 1 below, and to reviewer 1, major comment 5.

Remembering the wrong eccentricity: We focused our analyses on polar angle because this is the dimension on which our stimuli varied and the dimension for which pRF estimates are more reliable (Benson et al., 2018). However, we can also evaluate BOLD responses across eccentricity. We now briefly explore eccentricity-dependent responses during perception and

memory (Results, paragraph beginning line 225). In Supplementary Figure 1, we plot the average BOLD response from 0 to 8 degrees eccentricity, averaging across subjects, stimuli, and voxels with different polar angle preferences. In most regions, perception responses peak near the eccentricity that the stimulus was presented at, falling off sharply toward the fovea and slowly toward the periphery. During memory, we see similar peak locations, though with much shallower drop-offs, paralleling our polar angle results. Though these results are much noisier than polar angle results (especially beyond V3, as expected from Benson et al., 2018), we believe they are consistent with subjects remembering the stimulus at the correct eccentricity.

Supplementary Figure 1

Minor comments:

(1) Related to the above comment of the memory task being effectively categorical, I wanted to follow up on modeling the memory errors. The authors say they have to assume implausibly large errors (i.e. 45°) for this model to approach the memory data. However, what if subjects accidentally mix up two cues, and thus confuse two quadrants with each other. Such a misbinding of cue-to-quadrant would lead not to small errors, but to errors of any multitude of 45° .

Thank you for this suggestion, which was also made by Reviewer 1. We agree that it is reasonable to think that subjects might make these kinds of associative errors. To address this, we have added a new simulation to the paper (Results subsection “Associative Error Simulation”, beginning line 319), and we have added a new panel to Figure 5. Similar to our other simulations, very high error rates ($>50\%$) are required to artifactually produce something similar to the memory responses we observed with no underlying changes to the neural response properties. We think it’s very unlikely that subjects are making associative errors a majority of the time, as the proportion of such errors in the last round of training was 0%. We also find that, like the other simulations, this simulation cannot reproduce data from all ROIs

with the same set of parameters. That is, as we increase the number of associative errors and get closer to approximating the memory data in V1, our approximation of the memory response in V3ab gets worse (see Fig. 5c, Supplementary Fig. 3c, and Supplementary Fig. 4c).

(2) Line 74: For interpretation of results going forward, it would be useful to include #subjects here.

The text now reads: “Prior to being scanned, subjects (N=9) participated in a behavioral training session.” (Results, line 92)

(3) Figure 4a: For clarification, I assume the left two panels are a single subject, while the right most panel is the average across all subjects, correct?

Thank you for raising this point. We have revised this part of Figure 4. In our original submission, the leftmost panels were both a single subject, but the rightmost panel was a cartoon. We have reduced the left panels into one panel and revised the rightmost panel to show actual data for the example subject instead of a cartoon. In addition, we also now show group data with a von Mises fit. Whether the panels indicate single subject or group data is now clearly labeled in the caption and directly in the figure.

Figure 4a

(4) Figure 4: While it’s clear from the data that dark-orange is probably perception (and light orange is memory), a legend would be useful for the lazy reader. Especially since the figure legend does not clarify either – some potential for confusion.

Thank you. We have added a legend to prevent confusion.

(5) Line 146: “the fact the” should probably be “the fact that”?

We have corrected this typo.

(6) Are amplitude and FWHM truly independent in this fitting procedure? In the absence of a good intuition, is it possible to see some kind of scatter plot of the two plotted against each other with dots for each bootstrap.

Thank you for raising this important point. There are several different issues at hand here. The first is whether amplitude and FWHM are correlated in our data. Below, we plot the FWHM vs amplitude for perception and memory bootstraps. Each dot is a bootstrap and dots are colored by ROIs. During perception, there is a clear relationship between FWHM and amplitude across ROIs. ROIs with higher FWHM also have lower amplitude. In our hierarchical model, a single operation (convolution) generates both of these effects. In some ROIs, this relationship is also present across bootstraps. For instance, bootstraps with higher perception FWHM in LO also have lower amplitudes. This could indicate an inverse relationship between FWHM and amplitude across subjects, with low FWHM/high amplitude subjects contributing more to some bootstraps and high FWHM/low amplitude subjects contributing more to others, though we are hesitant to speculate about this too much with the small number of subjects. During memory, relationships between FWHM and amplitude are not as clearly observed.

A second, related, issue is whether our von Mises fitting procedure somehow imposes a correlation between amplitude and FWHM in our data, and whether it's possible to recover different patterns with our fitting procedure. We simulate examples of high FWHM/high amplitude responses (left plot, dots) and low FWHM/low amplitude responses (right plot, dots). We then applied the same von Mises fitting procedure we used on the data to these simulations. Lines indicate the best fit, which capture the data well. Thus, we are confident that our procedure could correctly capture a positive correlation between FWHM and amplitude if it was present in our data.

A third issue is whether there is necessarily an inverse relationship between FWHM and amplitude across ROIs in all stimulus sets. We think the answer is no, as it is easy to think of ways to break this relationship. For instance, presenting a ring stimulus should yield an infinitely wide response across all polar angles, breaking this relationship. Presenting a much lower contrast stimulus to V1 should decrease the amplitude of the BOLD response without dramatically changing the spatial spread of the response. For these reasons, we are cautious about making generalizations about the relationship between amplitude and FWHM on the basis of this stimulus set alone. This issue is also discussed in our response to Reviewer 1, comment 3.

(7) line 200: how is empirical SNR calculated?

We estimate empirical SNR by bootstrapping our GLMs. Instead of running our GLMs just once per subject, we run it 100 times, resampling runs with replacement. This gives us a distribution of parameter estimates for a given voxel rather than just one parameter. We define a voxel's SNR for a given stimulus and task condition (perception or memory) to be the median value of that distribution divided by the standard error. Overall, we find that the median of the distribution for memory is reliably much lower than the median of the distribution for perception, while the widths of the distribution are much more comparable. This results in lower SNR for the memory task. As an example, we plot parameter estimate distributions for two example voxels below:

We clarify our procedure in detail in the Methods section.

“We determined the amount of signal and noise actually observed for each vertex during perception and memory by examining bootstrapped parameter estimate distributions produced by GLMdenoise. We defined the median parameter estimate across bootstraps as the amount of signal and the standard error of this distribution as the amount of noise. To simulate new data for a vertex, we randomly drew a new parameter estimate from a normal distribution defined by the true signal value (median) and the noise value (SE) needed to produce the target SNR.” (Methods, lines 899-904)

(8) line 239-140: typo? Or is the claim really that subjects experiencing a small, variable amount of memory error explains the memory data?

This was a typo. The sentence now reads: “Like previous simulations, these simulations show that angular memory error is an unlikely explanation for our observed results.” (Results, lines 336-337)

(9) I am kind of surprised at the shape of the V1-3 lines in Fig 5b (right) for the FWHM plot. It seems that at both 0% and 100% retrieval failure, the proportion of simulations that fall within the memory CI should be 0. This because at 100% retrieval failure the 1D polar angle response function is presumably flat, and FWHM approaches inf, i.e. always > the memory CI. I also would have expected that there would be some sweet spot in between 0% and 100% failure where the proportion within the memory CI would be quite high. Probably, I just have a wrong intuition about the analysis, but why is this not an inverted U shape for V1-3? Supp fig 1b does not convince me that 100% failure curves are not flat.

Thank you for raising this important point. We agree that the percentages plotted in Figure 5 can be unintuitive in some cases. One important difference between the 0% and 100% retrieval failure conditions in Figure 5b is the amount of noise. In the 0% retrieval failure condition (which we now refer to as “0% task lapse”), we are essentially reproducing our perception data, which has high signal-to-noise. In the 100% retrieval failure/lapse condition, we are replacing all of this data with pure noise, which we make correlated across voxels. Thus, when we attempt to fit curves to the data in this condition, it will sometimes fit a random spike of noise in one bin because there is no other signal to pick up on. To illustrate this, below, we first plot the average simulated data for the 100% retrieval failure/task lapse condition. As noted by the Reviewer, the average % signal change across all simulations is (and should be) flat across polar angles. Each black dot represents the average simulated response across all simulations and each blue dot is the value from a single simulation.

Now, we plot a few individual simulations, their fitted von Mises, and the FWHM derived from the fit. The fit (and the FWHM) in these cases reflects only noise. Note also that FWHM is only a read-out of the x coordinates at half the maximum value. It does not consider negative peaks or portions of the curve that lie below zero at all, circumstances that are increasingly common under conditions of high noise (e.g. middle and right panel below).

Because 360 degrees is the maximum possible FWHM and because the fits often reflect noise, the average FWHM across the simulations will be less than 360 degrees. Indeed, if we plot the FWHM from each simulation, there is a wide spread across the 0-360 range, with some additional simulations that are not pictured failing to fit. A minority of those values fall within the memory confidence interval by chance. We have added Supplementary Figure 4, which plots every FWHM from a simulation with respect to the confidence interval to show this directly. We hope this aids readers in interpreting these simulations.

Supplementary Figure 4b

There are a couple of reasons there is no clear “peak” in Figure 5b for V1-V3. One reason could be that the sampling on the x-axis is not dense enough. It’s possible that an 80% lapse rate would achieve more simulations within the confidence intervals than 75% or 100%. We don’t believe it’s necessary to estimate the peak value in order to establish that this is an unlikely explanation of our main results. Second, we never expect for the majority of simulations to fall within the memory confidence intervals for an area like V1. In this simulation framework (adding noise to the perception data), the only way we can achieve *some* simulations falling into this range is by massively increasing the dispersion of values from the true perception value. A

different kind of simulation could shift the mean FWHM upward without increasing dispersion, but not the kind we have set out to perform here.

(10) Line 312 (and 406-409): Would it help to test this by collapsing early and later visual ROIs (i.e. “giant” early V1-3 ROI vs. “giant” late ROI hV4/LO/V3ab)? This might show a significant reversal.

This analysis tells the same story as the one we report in the paper. The interaction between early/late and perception/memory is highly significant ($\beta = 65.7$, 95% CI = [20.3, 116.7]). Within memory alone, the difference between early and late is not reliable ($\beta = -19.3$, 95% CI = [-60.54, 28.7]), though FWHM is numerically smaller for the late ROI than for the early ROI.

(11) With respect to the hierarchical model, how were kernel sizes chosen? did the authors try to fit the size of the convolutional kernel to the data? For example, if fit only to the perception data, how well can memory data be predicted from kernel sizes derived from fitting perception data? And when both perception and memory data are used to constrain the model?

Thank you for raising this point, which makes contact with a point made by Reviewer 1, comment 4, subpoint (ii). We agree that our analysis should be expanded and our approach better motivated.

We did not try to fit the size of the kernel to our perception or memory data. We are hesitant to do this for several reasons. First, the exact amplitude and FWHM values we report in the paper are very likely to depend on the exact stimuli we used. While we report a specific FWHM of 60° during perception and 100° during memory in V3, it is not clear whether the exact values would be the same for colorful shapes, faces, etc. Given this, our aim was to identify a class of models that would reproduce the general patterns we observed (e.g. maximally different perception FWHM and memory FWHM in the earliest layer) regardless of the exact parameters used and values produced. We think this is a more useful contribution than fitting the model to one experiment. We have edited this section of the Results to better motivate our approach:

“Importantly, we did not fit our model to our data. Nor did we attempt to generate a model that would predict the exact FWHM and amplitude values we report. Instead, we sought to identify a *class* of models where most parameter choices yielded the qualitative pattern of data we observed during both perception and memory.” (Results, lines 412-414)

In order to better support our claims, we have also added Supplementary Figure 7, which explores how the number of layers, stimulus size, and kernel size impact the hierarchical model FWHM. Importantly, while we can produce different FWHM values depending on the exact parameters used, and small or large versions of our effects depending on the exact parameters used, we can't reverse any of our key trends. We discuss this explicitly in the Results section:

“We note that while the exact FWHM values depend on the number of layers, the stimulus size, and the kernel size (Supplementary Fig. 7), the qualitative patterns we focus on are general properties of this model class and can be observed across many parameter choices. Though some parameter choices can result in little to no change in FWHM, there are no parameter choices that can reverse the trends we report -- for example, by producing *smaller* FWHM in layer 1 than in layer 2 during feedback.” (Results, line 444-448)

(12) Out of curiosity, did the authors explore the relationship between vividness ratings and 1D response functions?

This is an interesting question. However, there was not enough variance in the vividness ratings to support this analysis (89.8% of ratings were the highest rating “vivid”). We expected this to be the case in this paradigm due to the very small number of memories. Our motivation for having subjects make these ratings was for subjects to generate their own feedback about their retrieval success -- i.e. if they were evaluating many memories as vivid, they were successfully performing the task, and if they had a string of lower ratings, they needed to try harder to engage with the task. We agree if it would be interesting to explore this relationship in a study with more variability.

References not cited in main manuscript:

- Parkinson, J.K., Murray, E.A., & Mishkin, M. (1988). A selective mnemonic role for the hippocampus in monkeys: memory for the location of objects. *Journal of Neuroscience*, 8(11), 4159-4167.
- Gilbert, P. E., & Kesner, R. P. (2003). Localization of Function Within the Dorsal Hippocampus: The Role of the CA3 Subregion in Paired-Associate Learning. *Behavioral Neuroscience*, 117(6), 1385–1394.
- Nyberg, L., Habib, R., McIntosh, A.R., & Tulving, E. (2000). Reactivation of encoding-related brain activity during memory retrieval. *Proceedings of the National Academy of Sciences*, 97(20), 11120-11124.
- Markov, N.T..., Kennedy, H. (2013). Anatomy of hierarchy: Feedforward and feedback pathways in macaque visual cortex. *Journal of Comparative Neurology*, 522(1), 225-259.
- Tootell, R.B.H, & Hadjikhani, N. (2001). Where is ‘Dorsal V4’ in Human Visual Cortex? Retinotopic, Topographic and Functional Evidence. *Cerebral Cortex*, 11(4), 298–311.
- Van Essen, D.C., Lewis, J.W., Drury, H.A., Hadjikhani, N., Tootell, R.B.H., Bakircioglu, M., & Miller, M.I. (2001). Mapping visual cortex in monkeys and humans using surface-based atlases. *Vision Research*, 41 (10-11), 1359-1378.
- Mishkin, M., Ungerleider, L.G., & Macko, K.A. (1983). Object vision and spatial vision: two cortical pathways. *Trends in Neurosciences*, 6, 414-417.
- Jenison, A., Wixted, J.T., Hopkins, R.O., & Squire, L.R. (2012). Visual Working Memory Capacity and the Medial Temporal Lobe. *Journal of Neuroscience*, 32(10), 3584-3589.

Reviewers' Comments:

Reviewer #1:

Remarks to the Author:

The authors present a thorough revision of their manuscript, including many helpful additional analyses and clarifications. These changes address all my previous points except for one, concerning the classification of the task as hippocampus-dependent (previous point #6).

In this previous comment, I asked if the authors could provide empirical evidence to support their claim that the task is hippocampally dependent. For example, previous work has shown that (univariate) engagement of the hippocampus is related to the degree of neocortical reinstatement. A conceptually similar analysis could be done here to establish a relationship between hippocampal engagement and the precision of the memory-related fMRI responses. In their rebuttal, the authors argue that there is much theoretical reason to believe that the hippocampus is involved in paired associate tasks, which I agree with in principle. Showing such involvement empirically may not be critical for the conclusions of this study but would help clarify the nature of the task used here (see also the other reviewers' points regarding imagery, attention to spatial locations, and working memory).

The editor also asked explicitly if the novelty concern (previous point #1) has been sufficiently addressed. The manuscript now discusses the overlap and differences with the study by Breedlove and colleagues (2020, *CurrBiol*) in some detail. In my opinion, the studies come to similar conclusions from different angles, and thus complement each other well.

Reviewer #2:

Remarks to the Author:

The revised manuscript is thoughtfully written and the authors have been responsive to the queries and concerns raised in the previous round of reviews. A major concern about the original submission was the extent to which the findings should be considered interesting and novel given the extensive previous work showing how top-down effects of feedback in various cognitive tasks (attention, imagery, visual working memory) can modulate the responses of the early visual cortex. The authors point out how from informal inspection, the visual working memory literature shows that memory for spatial location is not that different from the response profiles observed with perception, and certainly those memory-based profiles appear much sharper than those reported in this study regarding long-term memory reactivation. They also note how their reported findings deviate somewhat from the predictions of the recent paper by Breedlove et al.

In reading this resubmitted manuscript, I feel that the paper does provide a substantive contribution and to me, there is suggestive evidence of differences in the spatial precision of top-down feedback resulting from long-term memory vs. working memory. Also, it is not straightforward to show that the apparent loss of precision in the memory reactivation task is not due to the generally lower amplitude of BOLD responses found in individual visual areas. Overall, this is a rigorous study, the discussion section has been considerably expanded and provides a much broader context for appreciating the study's findings, and I feel convinced of the main empirical results. For these reasons, I recommend publication.

Reviewer #4:

Remarks to the Author:

I was asked to assess if the authors' reply to reviewer #3 adequately addressed all of that reviewer's concerns. The authors provided thorough and convincing replies to each of the reviewer's concerns,

presenting the results of new analyses and simulations, where needed.

Reviewer #3's major concern was that some of the results could be explained by subject error and / or lack of compliance. This is always a risk with retrieval / imagery studies, but I think the authors' approach of simulating the effects of specific kinds of memory errors is laudable, convincing, and adds a lot of value to the work.

We thank the reviewers for their positive review and for all their helpful feedback. We have addressed the remaining point by Reviewer 1 below.

Reviewer #1 (Remarks to the Author):

The authors present a thorough revision of their manuscript, including many helpful additional analyses and clarifications. These changes address all my previous points except for one, concerning the classification of the task as hippocampus-dependent (previous point #6).

In this previous comment, I asked if the authors could provide empirical evidence to support their claim that the task is hippocampally dependent. For example, previous work has shown that (univariate) engagement of the hippocampus is related to the degree of neocortical reinstatement. A conceptually similar analysis could be done here to establish a relationship between hippocampal engagement and the precision of the memory-related fMRI responses. In their rebuttal, the authors argue that there is much theoretical reason to believe that the hippocampus is involved in paired associate tasks, which I agree with in principle. Showing such involvement empirically may not be critical for the conclusions of this study but would help clarify the nature of the task used here (see also the other reviewers' points regarding imagery, attention to spatial locations, and working memory).

The editor also asked explicitly if the novelty concern (previous point #1) has been sufficiently addressed. The manuscript now discusses the overlap and differences with the study by Breedlove and colleagues (2020, *CurrBiol*) in some detail. In my opinion, the studies come to similar conclusions from different angles, and thus complement each other well.

We agree with the Reviewer that considering whether the hippocampus is involved in this task is useful in interpreting the data. Following the Reviewer's suggestion, we show below that there is an evoked response in the hippocampus during our memory retrieval task, peaking on average 7 seconds after the onset of the retrieval cue, with variability in the timing and size of the response across participants. As a reminder, the only perceptual change in this task is a .1 degree fixation dot changing hue; thus we interpret responses in the hippocampus to more likely reflect retrieval than visual stimulation. We believe this analysis is consistent with our perspective that the hippocampus is providing some top-down drive to visual cortex in this task. While we have not added this analysis to our manuscript, we are willing to do so if the reviewer requests it. We would also like to emphasize that hippocampal involvement in the retrieval task is a point of view we take when discussing our results about visual cortex (with the support of prior papers, and now, this new analysis), but that no conclusions about the hippocampus are made by our paper and that the term "hippocampus" does not appear in our title, abstract, or introduction.

Average evoked signal across all voxels in the bilateral hippocampus during memory retrieval, estimated with an FIR model and averaged across nine participants. Shaded areas indicate SE of the mean across participants.

Average evoked signal across all voxels in the bilateral hippocampus during memory retrieval (as above), plotted separately for each of the nine participants.

We share the Reviewer’s interest in understanding the relationship between hippocampal activation and the precision of the visual cortical responses we report. However, as we noted in the prior response, we do not think this dataset provides a strong test of the hypothesis that

hippocampal activation and visual cortical precision are correlated on a trial-by-trial basis. This is because we intentionally eliminated as much variance in memory precision as possible, in order to test the hypothesis that there are differences between perception and memory activation that are not due to memory weakness/failure. This choice makes it difficult to then perform analyses that presuppose a lot of variance in retrieval precision in every individual participant. While it would still be possible to perform this analysis anyway, we think it would be a weak test of the idea, and that it would not be a meaningful contribution to the literature regardless of the outcome. A future study designed specifically to test this idea should be conducted.

That said, we agree that our manuscript would benefit from some caveats around this point. Memory retrieval is not supported by the hippocampus alone. In the discussion, we now make it clear that other high-level cortical regions may also be a source of top-down drive to visual cortex. Imagery studies frequently rely on tasks that, in our view, are unlikely to be hippocampally-dependent (e.g. “Picture a tree”, “Imagine the letter A”). These tasks nevertheless show activation in visual cortex (e.g. Ganis, Thompson, & Kosslyn, *Cognitive Brain Research*, 2004). We now clarify that we don’t believe the hippocampus is necessary to observe visual cortex activation in the absence of visual stimulation, and that top-down drive from other cortical regions may achieve something similar. Indeed, as shown in examples below, we observe widespread activation in lateral parietal, prefrontal, and temporal cortex as well as medial areas during memory retrieval (as have many others), and it’s possible that these regions are driving the activation we report in visual cortex in addition to or instead of the hippocampus. We have adjusted the discussion of our paper to make this possibility more clear to the reader, though, as noted previously, this does not impact any of the conclusions of the paper.

Cortical activation during memory retrieval in two individual participants. Timeseries were smoothed, and then activation was estimated using a single regressor for all retrieval trials and contrasted versus baseline. The resulting maps were projected onto individual surface meshes.

Changes to manuscript:

Results, lines 406-407:

“Reinstated cortical activity is then thought to propagate backwards through visual cortex (Naya et al., 2001; Linde-Domingo et al., 2019; Dijkstra et al., 2019; Hindy et al., 2016), driven by the hippocampus and/or its cortical connections.”

Discussion, lines 549-559:

“Studies also show that the hippocampus sits atop the highest stage of the visual hierarchy, with reciprocal connections to high-level visual regions via the medial temporal lobe cortex (Van Hoesen & Pandya, 1975; Felleman & Essen, 1991; Suzuki & Amaral, 1994). The hippocampus is also strongly coupled to a widespread cortical network, including retrosplenial/posterior cingulate cortex, lateral parietal and temporal cortices, and lateral and medial prefrontal cortices (Rugg & Vilberg, 2013; Cooper & Ritchey, 2019). Some of these regions may also exert top-down drive over visual cortex. These observations make the prediction that initial top-down drive from the hippocampus and its cortical network during memory retrieval may result in the backwards propagation of neural activity through the visual system. Neural recordings from the macaque (Naya et al., 2001) and human (Hindy et al., 2016; Linde-Domingo et al., 2019; Dijkstra et al., 2019), as well as computational modeling (Horikawa & Kamitani, 2017) support the idea of backwards propagation.”

Discussion, lines 603-605:

“Long-term forms of episodic memory retrieval and imagery are both thought to be capable of driving visual activity at much longer delays. While episodic memory retrieval depends on the hippocampus, some forms of imagery may not require hippocampal processing at all (Rosenbaum et al., 2004).”

Discussion, lines 619-622:

“However, during episodic memory tasks such as the paired associates task we use here, there is no recently evoked representation of the stimulus in visual cortex. This representation must be created anew, and in standard models of memory retrieval, this process is initiated by the hippocampus and its cortical network.”

Reviewer #2 (Remarks to the Author):

The revised manuscript is thoughtfully written and the authors have been responsive to the queries and concerns raised in the previous round of reviews. A major concern about the original submission was the extent to which the findings should be considered interesting and novel given the extensive previous work showing how top-down effects of feedback in various cognitive tasks (attention, imagery, visual working memory) can modulate the responses of the early visual cortex. The authors point out how from informal inspection, the visual working memory literature shows that memory for spatial location is not that different from the response profiles observed with perception, and certainly those memory-based profiles appear much sharper than those reported in this study regarding long-term memory reactivation. They also

note how their reported findings deviate somewhat from the predictions of the recent paper by Breedlove et al.

In reading this resubmitted manuscript, I feel that the paper does provide a substantive contribution and to me, there is suggestive evidence of differences in the spatial precision of top-down feedback resulting from long-term memory vs. working memory. Also, it is not straightforward to show that the apparent loss of precision in the memory reactivation task is not due to the generally lower amplitude of BOLD responses found in individual visual areas. Overall, this is a rigorous study, the discussion section has been considerably expanded and provides a much broader context for appreciating the study's findings, and I feel convinced of the main empirical results. For these reasons, I recommend publication.

We thank the reviewer for their thoughtful comments on the paper.

Reviewer #4 (Remarks to the Author):

I was asked to assess if the authors' reply to reviewer #3 adequately addressed all of that reviewer's concerns. The authors provided thorough and convincing replies to each of the reviewer's concerns, presenting the results of new analyses and simulations, where needed.

Reviewer #3's major concern was that some of the results could be explained by subject error and / or lack of compliance. This is always a risk with retrieval / imagery studies, but I think the authors' approach of simulating the effects of specific kinds of memory errors is laudable, convincing, and adds a lot of value to the work.

We thank the reviewer for their careful review of our revisions.

Reviewers' Comments:

Reviewer #1:

Remarks to the Author:

The authors adequately addressed the one remaining comment in their latest revision, regarding the contribution of the hippocampus to this task. They decided not to conduct a trial-by-trial analysis relating hippocampal activity to visual cortex representations. This is fine, since the central conclusions do not depend on the task being hippocampus-dependent. I do not believe the analyses presented in the response letter would add much value to the manuscript, and the discussion is now sufficiently clear about the possible generators of top-down signals. Hence from my view, the manuscript is ready for publication, and I congratulate the authors on this nice work.